# Evidence of strong and mode-selective electron–phonon coupling in the topological superconductor candidate 2M-WS$_2$

Yiwei Li [1,12] ✉, Lixuan Xu [2,12], Gan Liu [3,4,12], Yuqiang Fang [5,12], Huijun Zheng [6,7], Shenghao Dai [1], Enting Li [1], Guang Zhu [1], Shihao Zhang [8], Shiheng Liang [2], Lexian Yang [9], Fuqiang Huang [5], Xiaoxiang Xi [3,4] ✉, Zhongkai Liu [6,7] ✉, Nan Xu [1,10] ✉ & Yulin Chen [6,7,11]

The interaction between lattice vibrations and electrons plays a key role in various aspects of condensed matter physics – including electron hydrodynamics, strange metal behavior, and high-temperature superconductivity. In this study, we present systematic investigations using Raman scattering and angle-resolved photoemission spectroscopy (ARPES) to examine the phononic and electronic subsystems of the topological superconductor candidate 2M-WS$_2$. Raman scattering exhibits an anomalous nonmonotonic temperature dependence of phonon linewidths, indicative of strong phonon–electron scattering over phonon–phonon scattering. The ARPES results demonstrate pronounced dispersion anomalies (kinks) at multiple binding energies within both bulk and topological surface states, indicating a robust and mode-selective coupling between the electronic states and various phonon modes. These experimental findings align with previous calculations of the Eliashberg function, providing a deeper understanding of the highest superconducting transition temperature observed in 2M-WS$_2$ (8.8 K) among all transition metal dichalcogenides as induced by electron–phonon coupling. Furthermore, our results may offer valuable insights into other properties of 2M-WS$_2$ and guide the search for high-temperature topological superconductors.

The lattice and electrons are two integral subsystems of crystalline materials, and their interplay significantly influences various properties[1], such as electrical resistivity of metals[2], carrier mobility of semiconductors[3], and thermoelectric behaviors[4]. Furthermore, a recent accumulation of theoretical and experimental studies highlights the substantial involvement of electron–phonon coupling (EPC) in the formation of numerous exotic quantum states, including electron–phonon liquids[5,6], charge-ordered states[7,8], high-temperature

[1]Institute for Advanced Studies (IAS), Wuhan University, Wuhan, China. [2]Department of Physics, Hubei University, Wuhan, China. [3]National Laboratory of Solid State Microstructures and Department of Physics, Nanjing University, Nanjing, China. [4]Collaborative Innovation Center of Advanced Microstructures, Nanjing University, Nanjing, China. [5]School of Materials Science and Engineering, Shanghai Jiao Tong University, Shanghai, China. [6]School of Physical Science and Technology, ShanghaiTech University, Shanghai, China. [7]ShanghaiTech Laboratory for Topological Physics, Shanghai, China. [8]School of Physics and Electronics, Hunan University, Changsha, China. [9]State Key Laboratory of Low Dimensional Quantum Physics, Department of Physics, Tsinghua University, Beijing, China. [10]Wuhan Institute of Quantum Technology, Wuhan, China. [11]Department of Physics, University of Oxford, Oxford, UK. [12]These authors contributed equally: Yiwei Li, Lixuan Xu, Gan Liu, Yuqiang Fang. ✉e-mail: yiweili@whu.edu.cn; xxi@nju.edu.cn; liuzhk@shanghaitech.edu.cn; nxu@whu.edu.cn

superconductors[9,10], and strange metals[11,12], in addition to its fundamental role in the conventional Bardeen–Cooper–Schrieffer (BCS) superconductivity[13].

2M-WS$_2$ has recently been theoretically proposed as an intrinsic topological superconductor candidate, and it exhibits a superconducting transition temperature $T_C$ = 8.8 K – the highest among all stoichiometric transition metal dichalcogenides (TMDs) under ambient pressure[14]. Experimentally, angle-resolved photoemission spectroscopy (ARPES) investigation has observed topological surface states[15,16] and scanning tunneling microscopy/spectroscopy (STM/STS) study has discovered signatures of zero Majorana modes at its magnetic vortex cores[17], both demonstrating nontrivial band structure topology. In addition, 2M-WS$_2$ has exhibited a rich phase diagram, featuring Pauli-limit violated superconductivity[18,19], surface charge-ordered states[20], and strange metal behavior[21]. However, the physical mechanism underlying is far from understood. Strong EPC in 2M-WS$_2$, as suggested by the first-principles calculations based on Migdal–Eliashberg theory[22], might contribute to its relatively high $T_C$ and play an essential part in the emergence of other exotic phases. Nevertheless, there is currently a lack of direct experimental evidence supporting this hypothesis.

In this study, the phononic and electronic subsystems of 2M-WS$_2$ are investigated by Raman scattering and ARPES, respectively, showing unambiguous experimental evidence of EPC-induced renormalized phononic and electronic band structures. Remarkably, the Raman scattering reveals an unusual nonmonotonic temperature dependence of phonon linewidths for selective modes. This observation is interpreted as a prevalence of strong phonon–electron scattering over phonon–phonon scattering as the temperature decreases[5,6]. In addition, low-energy electronic dispersions are heavily distorted due to strong EPC, resulting in kinks at multiple binding energies discovered by high-resolution laser-ARPES. The strength of mode-selective EPC is

quantitatively assessed by fitting the temperature-dependent phonon linewidths obtained from Raman scattering measurements and through careful self-energy analysis of ARPES spectra. The experimental estimated EPC coupling strength is in general agreement with the first-principles calculated Eliashberg function[22]. Our findings of mode-selective EPC not only establish a benchmark for theoretical modeling of various exotic phases observed in 2M-WS$_2$ but also offer insight into the quest for high-$T_C$ topological superconductors. As a paradigm experimental investigation on EPC, we expect that this methodology can also be extended to other correlated materials.

## Results

### General physical properties

2M-WS$_2$ crystallizes in a centrosymmetric base-centered monoclinic structure (space group C2/m, No. 12)[14]. The framework of W atoms is illustrated in Fig. 1a with the conventional and primitive unit cells indicated by the blue and magenta parallelepipeds, respectively. Within each layer (defined as $b$-$c$ plane), the displacement of W atoms from the 1T'-structure [WS$_6$]$^{8-}$ octahedral center results in a zigzag chain along the $b$-axis (see Fig. 1a). Figure 1b presents the bulk Brillouin zone (BZ) and (100) surface BZ marked with time-reversal momenta.

2M-WS$_2$ has recently attracted research enthusiasm since it is an intrinsic topological superconductor candidate hosting a rich phase diagram, as schematically illustrated in Fig. 1c, d. It undergoes a crossover from the Fermi liquid state to the strange metal state near 25 K and enters a bad metal state above 200 K, as identified by temperature-dependent electrical and thermal transport measurements (Fig. 1c)[21]. Both n-type and p-type doped samples exhibit suppression of $T_C$, displaying a typical superconducting dome with the intrinsic 2M-WS$_2$ at the optimal point ($T_C$ = 8.8 K) in the phase diagram of $T_C$ and carrier concentration (Fig. 1d)[23,24].

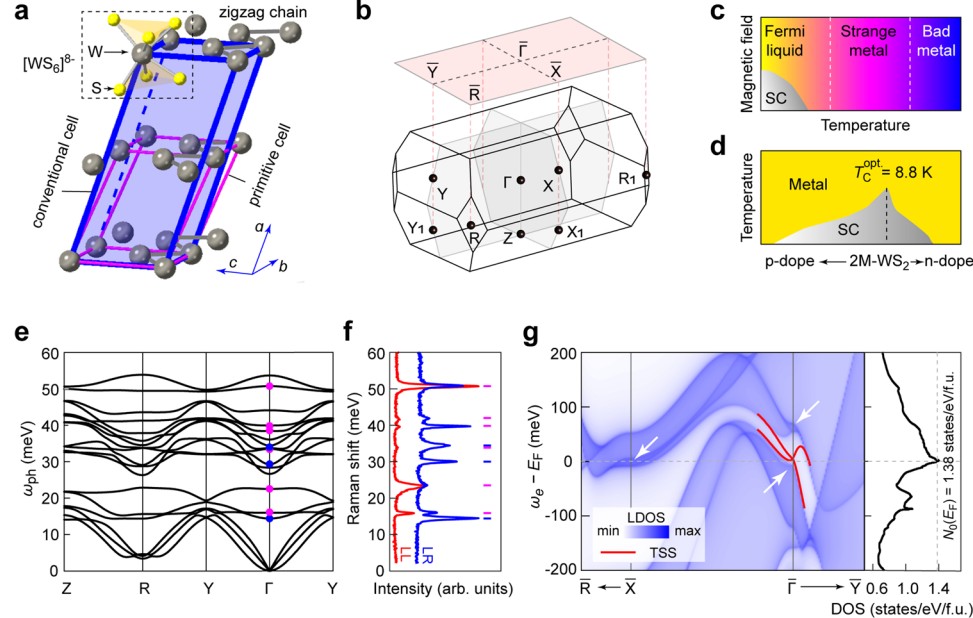

**Fig. 1 | General information of 2M-WS$_2$. a** Schematic illustration of the crystal structure of 2M-WS$_2$. Only the framework of W atoms is shown. The inset shows the building block structure of [WS$_6$]$^{8-}$. The boundaries of the primitive and conventional unit cells are indicated by magenta and blue lines, respectively. **b** Bulk Brillouin zone (BZ) marked with eight time-reversal-invariant momenta. The red rectangle plane indicates the 2D BZ of the (100) natural cleavage surface. **c** Schematic phase diagram as a function of temperature and magnetic field[21]. **d** Schematic phase diagram as a function of carrier concentration and temperature[24]. $T_C^{opt.}$ is the optimal superconducting transition temperature. **e** First-principles calculated phonon dispersions. The magenta and blue dots mark the

Raman active $A_g$ and $B_g$ modes, respectively. **f** Raman spectra measured at 5 K in circularly co-polarization (LL) and cross-polarization (LR) configurations, respectively. LL (LR): the first letter represents the polarized state of excited light, whereas the second letter represents the polarized state of scattering Raman signal. L and R denote left-handed and right-handed circularly polarized light, respectively. **g** Left panel: first-principles calculated (100)-surface-projected noninteracting band structure. The topological surface states (TSSs) are appended with red lines. The white arrows indicate the saddle-like bands near $E_F$. Right panel: The total noninteracting bulk density of states (DOS) peaks at $E_F$ with the peak value of 1.38 states/eV/f.u. (f.u. represents formula unit).

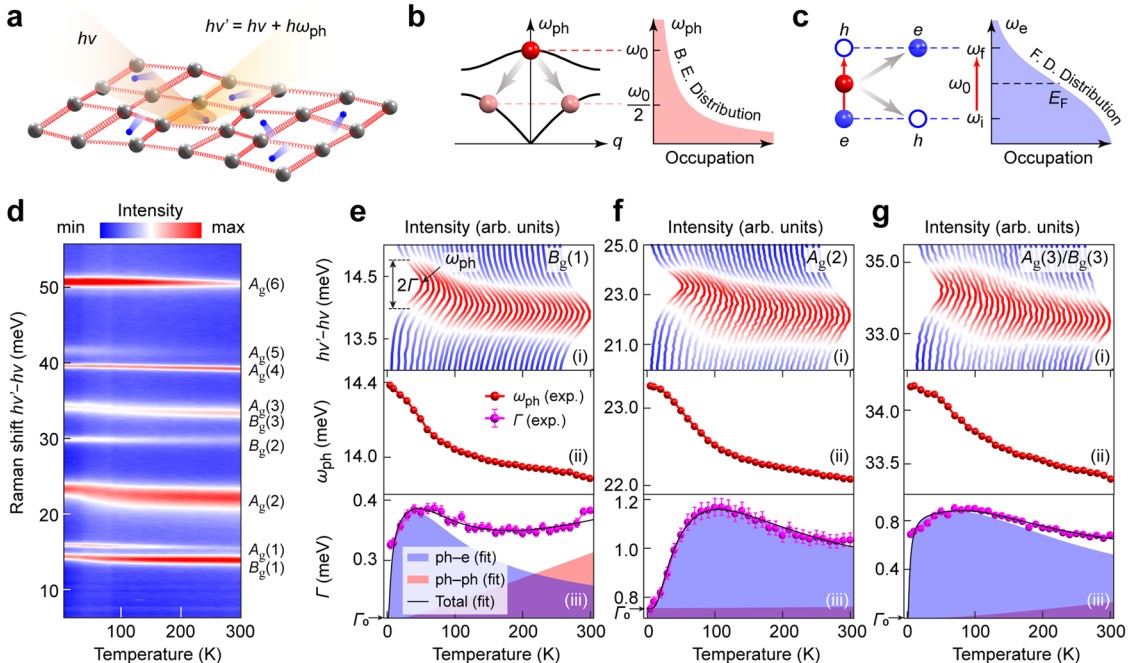

**Fig. 2 | Temperature-dependent Raman spectroscopy measurements.**
**a** Schematic illustration of the Raman scattering. Blue balls represent electrons. Gray balls connected with red springs represent the collective elastic arrangement of the lattice, phonons. $h\nu$ and $h\nu'$ are energies of the incident and scattered photons, respectively. $h\omega_{ph}$ is the phonon energy. **b** Schematic diagram showing the lowest-order anharmonic decay of an optical phonon with energy $\omega$ decaying into two acoustic phonons with energy $\omega/2$. The phonon distribution obeys Bose-Einstein (B. E.) statistics. **c** Schematic diagram showing an optical phonon with energy $\omega_0$ decaying into an electron-hole pair with energy interval $\omega_0$. The red balls represent phonons. The solid and hollow blue balls represent electrons and holes, respectively. The electron distribution obeys Fermi–Dirac (F. D.) statistics. **d** Temperature-dependent Raman spectra from 5 to 300 K. The $A_g(3)$ and $B_g(3)$

modes are indiscernible with a small energy difference of ~0.3 meV. **e** Top panel: stacking plots of temperature-dependent Raman spectra of $B_g(1)$ mode. Middle panel: Fitted temperature-dependent phonon energy of $B_g(1)$ mode based on Lorentzian functions. Bottom panel: Fitted temperature-dependent phonon linewidth (half-width-at-half-maximum of the Lorentzian profile) of $B_g(1)$ mode interpreted by combined contributions from phonon–electron scattering (blue shaded area) and phonon–phonon scattering (red shaded area) based on Eq. (1). The fitted results from these two scattering mechanisms are upshifted by a constant background $\Gamma_0$ (as indicated by black arrows) for a better illustration. **f, g,** Same as (**e**) but for $A_g(2)$ and $A_g(3)/B_g(3)$, respectively. Error bars are standard deviations obtained from the Lorentzian fits to the phonon peaks.

The first-principles calculated bulk phonon spectrum reveals 3 acoustic and 15 optical modes of 2M-WS$_2$ (Fig. 1e). This arises from each primitive unit cell containing 6 atoms (W$_2$S$_4$), thus resulting in 18 degrees of freedom (see Fig. 1a). Among these 15 optical modes, 9 of them are Raman active with the irreducible representations $\Gamma_{opt.}$ = $6A_g + 3B_g$, as marked by the magenta and blue dots (Fig. 1e). The Raman spectra (Fig. 1f) confirm these 9 optical modes by polarization-dependent measurements, as $A_g$-modes are observable in both circularly co-polarization (LL) and cross-polarization (LR) configurations while $B_g$-modes only appear in LR configuration based on selection rules. These mode assignments are further verified by our full polarization angle-dependent Raman spectroscopy measurements (Supplementary Fig. 1).

Figure 1g presents the first-principles calculated noninteracting bulk electronic local density of states (LDOS) projected on the natural (100) surface appended with the topological surface states (TSSs, as highlighted by the red lines) near the Fermi level ($E_F$). In great contrast to conventional topological insulators, such as Bi$_2$Se$_3$[25] and Bi$_2$Te$_3$[26], where the TSSs emerge in the global bulk band gap with zero bulk DOS, the TSSs in 2M-WS$_2$ reside in a local bulk band gap with significant bulk DOS. The valence and conduction bands are strongly anisotropic and form saddle-like dispersions near $E_F$ around $\bar{\Gamma}$ and $\bar{X}$ (as indicated by the white arrows), giving rise to a DOS peak at $E_F$ (the noninteracting single-particle DOS $N_0(E_F) = 1.38$ states/eV/f.u., f.u. represents formula unit). This value, however, is significantly lower than that derived from the Sommerfeld coefficient measured by the temperature dependence of the specific heat[27] (the renormalized many-body interacting $N'(E_F) = 3.81$ states/eV/f.u. For the derivation of $N'(E_F)$ from the

Sommerfeld coefficient, please refer to Supplementary Table 3). The discrepancy, therefore, indicates that the low-energy electronic band dispersions are highly renormalized by many-particle correlations, such as electron correlations and EPC.

## Raman spectroscopy

We have carried out a systematic Raman spectroscopy investigation on the phononic degrees of freedom of 2M-WS$_2$ due to its high energy resolution and sensitivity to different scattering processes involving phonons, as illustrated in Fig. 2a. Anharmonic phonon–phonon decay and phonon–electron scattering are two main channels of optical phonon decay, as schematically plotted in Fig. 2b, c, respectively. In the lowest-order anharmonic process, an optical phonon of energy $\omega_0$ decays into two acoustic phonons of energy $\omega_0/2$ and opposite momenta (Fig. 2b). The phonon–electron scattering describes a process in which an optical phonon decays into an electron-hole pair via phonon-mediated electron excitation (Fig. 2c). The combination of these two mechanisms contributes to the finite phonon lifetime (and consequently, a finite linewidth); however, they exhibit distinct temperature-dependent behaviors[5,6].

To precisely determine the quantitative behavior of the phonon energies and linewidths, we have performed temperature-dependent Raman spectra of all observable modes from 5 to 300 K, as presented in Fig. 2d. Notably, three representative modes, $B_g(1)$, $A_g(2)$, and $A_g(3)/B_g(3)$, show drastically increased phonon energies as the temperature decreases below 100 K and exhibit unusual nonmonotonic temperature-dependent linewidths by fitting each mode with a Lorentzian profile, as shown in Fig. 2e–g. This observation is in great

contrast to the Klemens model[28] describing anharmonic phonon–phonon decay. Due to the bosonic nature of phonons, the temperature dependence of phonon–phonon decay is governed by the Bose-Einstein distribution function (see Fig. 2b). Consequently, the linewidth resulting from this origin must increase monotonically with temperature. However, by introducing phonon–electron scattering guided by the Fermi–Dirac distribution function (see Fig. 2c), the observed nonmonotonic temperature-dependent linewidths can be perfectly fitted by the following formula

$$\Gamma(T) = \Gamma_0 + \frac{2}{\exp\left(\frac{\omega_0}{2k_B T}\right) - 1}\Gamma_{ph-ph} + \left[\frac{1}{\exp\left(\frac{\omega_e}{k_B T}\right) + 1} - \frac{1}{\exp\left(\frac{\omega_e + \omega_0}{k_B T}\right) + 1}\right]\Gamma_{ph-e}$$

$$(1)$$

containing three terms – a temperature non-dependent background $\Gamma_0$ (due to trivial impurity scattering, boundary scattering, etc), a term due to anharmonic phonon–phonon decay that increases monotonically with temperature, and a term of phonon–electron scattering that exhibits nonmonotonic temperature-dependent behavior. $\omega_0$ is the corresponding optical phonon energy, $\omega_e$ is the energy of the electron's initial state with respect to the Fermi energy $E_F$, $k_B$ is the Boltzmann constant, $\Gamma_{ph-ph}$ and $\Gamma_{ph-e}$ indicate the magnitudes of these two mechanisms that contribute to the total phonon linewidth[5,6]. The physical description of the phonon–electron scattering involves a transition from the electron initial state to its final state through the absorption of a phonon, or in other words, the decay of a phonon into an electron-hole pair, as illustrated in Fig. 2c. The energy dependence of the electronic DOS is neglected in the phonon–electron scattering term since the Fermi–Dirac distribution function shows much more significant energy dependence in the temperature region (~ 100 K with $k_B T \sim$ 8 meV) and energy region ($\omega_e$ and $\omega_e + \omega_0$ are roughly in the energy window $E_F \pm 50$ meV) of interest. This phenomenological model does not rely on ab initio calculations, but it can well reproduce the temperature-dependent phonon linewidths for all observed Raman modes, as presented in the bottom panel of Fig. 2e–g and Supplementary Fig. 2.

The fitted results of our Raman spectroscopy measurements are summarized in Table 1 and Supplementary Table 2, showing nice consistent phonon energies with our first-principles calculations. Specifically, we focus on the fitted $\Gamma^i_{ph-e}$ of each mode $i$ by using Eq. (1), which is closely related to the mode-resolved EPC strength $\lambda^i_{e-ph}$ by[29–31]

$$\lambda_{e-ph} = \sum_i \lambda^i_{e-ph} = \sum_i \frac{4\Gamma^i_{ph-e}}{\pi N_0(E_F)\omega_i^2}$$

$$(2)$$

where $N_0(E_F) = 1.38$ states/eV/f.u. is the noninteracting electronic DOS at $E_F$ obtained from the ab initio calculation (see Fig. 1g) and $\omega_i$ is the phonon energy at zero temperature, which is extracted from the Raman measurements at the lowest temperature (5 K) by fitting to a Lorentzian function. The summation of all Raman active modes gives rise to a rough estimation of the total EPC constant $\lambda_{e-ph}$ of order 1 (3.52). Noticeably, $B_g(1)$, $A_g(2)$, and $A_g(3)/B_g(3)$ modes with energies of 14.4, 23.3, and 34.2 meV showing the most pronounced nonmonotonic temperature-dependent linewidths are the three modes with the highest EPC strength, contributing about 75% of $\lambda^{Raman}_{e-ph}$. Raman spectroscopy investigation on a different sample shows consistent results (Supplementary Table 2).

Raman scattering measurements have limitations in the accurate estimation of the EPC constant. In general, only the order of magnitude of the EPC constant can be reliably obtained[30,32]. The inaccuracy has complicated origins. In our case, we argue that Eq. (1) is not a first-principles result but merely based on a simple phenomenological model. For the phonon–phonon scattering, only the lowest order of anharmonicity is included. For the phonon–electron scattering, it does not rely on phoninic or electronic structures, as well as their temperature dependence. Moreover, Eq. (2) can become inaccurate due to possible strong electron correlation in 2M-WS$_2$, which is excluded in its theoretical framework[31]. Other factors, including unstable fitting procedures and improper choice of background $\Gamma_0$, are discussed in Supplementary Note 1. Although being a rough estimation, the order of magnitude of the total EPC constant $\lambda_{e-ph}$ is consistent with the calculated value (0.79)[22].

## Laser-ARPES

As a complementary approach, we have performed high-resolution laser-ARPES to search for signatures of EPC on the low-energy electronic band dispersions of 2M-WS$_2$, as illustrated in Fig. 3a. The Fermi surface mapping measurement on the natural cleaved (100) surface (Fig. 3b) shows highly anisotropic Fermi energy contours of TSSs and bulk states (BSs), consistent with previous studies[15,16]. The ARPES dispersion measurement cutting through the TSS and the BS (Fig. 3c) along the $\bar{Y} - \bar{\Gamma}$ direction manifests pronounced kinks at binding energies around 10 and 30 meV. These bosonic modes coupled with electronic states are attributed to phonons, as there is no magnon excitation in 2M-WS$_2$. The bulk band dispersion $\omega_e - k_y$ and the energy-dependent linewidth [half-width-at-half-maximum $W(\omega_e)$] are further extracted by fitting the momentum distribution curves (MDCs) to Lorentzian functions, as displayed in Fig. 3d, e, respectively. The energies of the phonons can be estimated by the peak positions of the second-order derivatives of the dispersion relation $k_y(\omega_e)$ or the peak positions of the first-order derivatives of the linewidth $W(\omega_e)$, because of a sudden decrease of the carrier's lifetime[33,34].

In Fig. 3f, the ARPES measured band dispersion of the TSS hosts a significantly reduced Fermi velocity $v_F^*$ compared to the first-principles calculated single-particle bare band Fermi velocity $v_F^0$, with the total renormalization factor being $\lambda^{v_F}_{tot.} = \frac{v_F^0}{v_F^*} - 1 \approx 1.9$. This EPC strength for a specific band shows a nicely consistent value with that derived from the renormalization of DOS at $E_F$ ($\lambda^{DOS}_{tot.} = \frac{N'(E_F)}{N_0(E_F)} - 1 = 1.76$, where $N_0(E_F)$ and $N'(E_F)$ are the noninteracting single-particle band DOS at $E_F$ derived from the first-principles calculation and the renormalized DOS derived from the temperature-dependent specific heat measurement[27]), both suggestive of correlated many-particle physics in 2M-WS$_2$. This coincidence implies that the momentum dependence of the many-body interactions in 2M-WS$_2$ might be weak. Although the superconductivity in 2M-WS$_2$ might not be perfectly isotropic[17,35], the anisotropy of the EPC and superconducting gap can be weak[15,22]. Detailed discussions on the momentum-dependent Fermi velocity renormalization can be found in Supplementary Note 2.

Similar to the observation of the BS, the kinks with binding energies of around 10 and 30 meV are also seen in the dispersion of the TSS, as indicated by the steplike enhancement of the imaginary part of the self-energy Im$\Sigma$ (Fig. 3g) and the peaks of the real part of the self-energy Re$\Sigma$ (Fig. 3h). These kink features are reproducible in multiple samples (Supplementary Fig. 3) and exhibit negligible momentum-dependence (Supplementary Fig. 4). The EPC strength can be extracted from the steplike increase of the imaginary part of the self-energy, $\Delta$Im$\Sigma$, as ref. 36

$$\lambda_{e-ph} = \frac{2}{\pi}\sum_i \frac{(\Delta\text{Im}\Sigma)^i}{\omega^i_{ph}} \approx 1.1.$$

$$(3)$$

Here, $\omega^i_{ph}$ are the phonon energies which correspond to the kink energies, with $\omega^1_{ph} \approx 10$ meV and $\omega^2_{ph} \approx 30$ meV. The steplike increases of the imaginary part of the self-energy, $(\Delta\text{Im}\Sigma)^i$, are estimated to be $(\Delta\text{Im}\Sigma)^1 \approx 9.7$ meV and $(\Delta\text{Im}\Sigma)^2 \approx 24.9$ meV between adjacent plateaus. The estimated EPC coupling strength ($\lambda_{e-ph} \approx 1.1$) is in overall agreement with the first-principles calculation ($\lambda^{cal.}_{e-ph} = 0.79$)[22]. Since the

**Table 1 | Electron–phonon coupling parameters extracted from the temperature-dependent Raman spectroscopy measurement**

| Peak | $B_g(1)$ | $A_g(1)$ | $A_g(2)$ | $B_g(2)$ | $A_g(3)/\,B_g(3)$ | $A_g(4)$ | $A_g(5)$ | $A_g(6)$ |
|---|---|---|---|---|---|---|---|---|
| $\omega_0^{cal.}$ (meV) | 14.4 | 16.0 | 22.5 | 29.3 | 33.5/33.8 | 38.4 | 39.9 | 50.7 |
| $\omega_0^{Raman.}$ (meV) | 14.4 | 15.9 | 23.3 | 29.9 | 34.2 | 39.6 | 41.7 | 50.8 |
| $\Gamma_{ph-e}^i$ (meV) | 0.41 | 0.24 | 1.23 | 0.28 | 1.82 | 0.47 | 0.16 | 0.76 |
| $\lambda_{e-ph}^i$ | 0.912 | 0.438 | 1.045 | 0.145 | 0.718 | 0.138 | 0.042 | 0.136 |

$\omega_0^{cal.}$ is the first-principles calculated phonon energies. $\omega_0^{Raman.}$ is the phonon energy extracted from the lowest-temperature (5 K) Raman measurement by Lorentzian fitting. $\Gamma_{ph-e}^i$ is the magnitude of mode-selective phonon linewidths due to phonon–electron scattering, which is obtained by fitting the temperature-dependent Raman phonon linewidth (see Fig. 2d–g) using Eq. (1) for each mode. $\lambda_{e-ph}^i$ is the mode-selective EPC strength derived from Eq. (2), where $N_0(E_F) = 1.38$ states/eV/f.u. = 2.76 states/eV/u.c. is obtained from the noninteracting ab initio calculation (see Fig. 1g). f.u. represents a formula unit, and u.c. represents a unit cell.

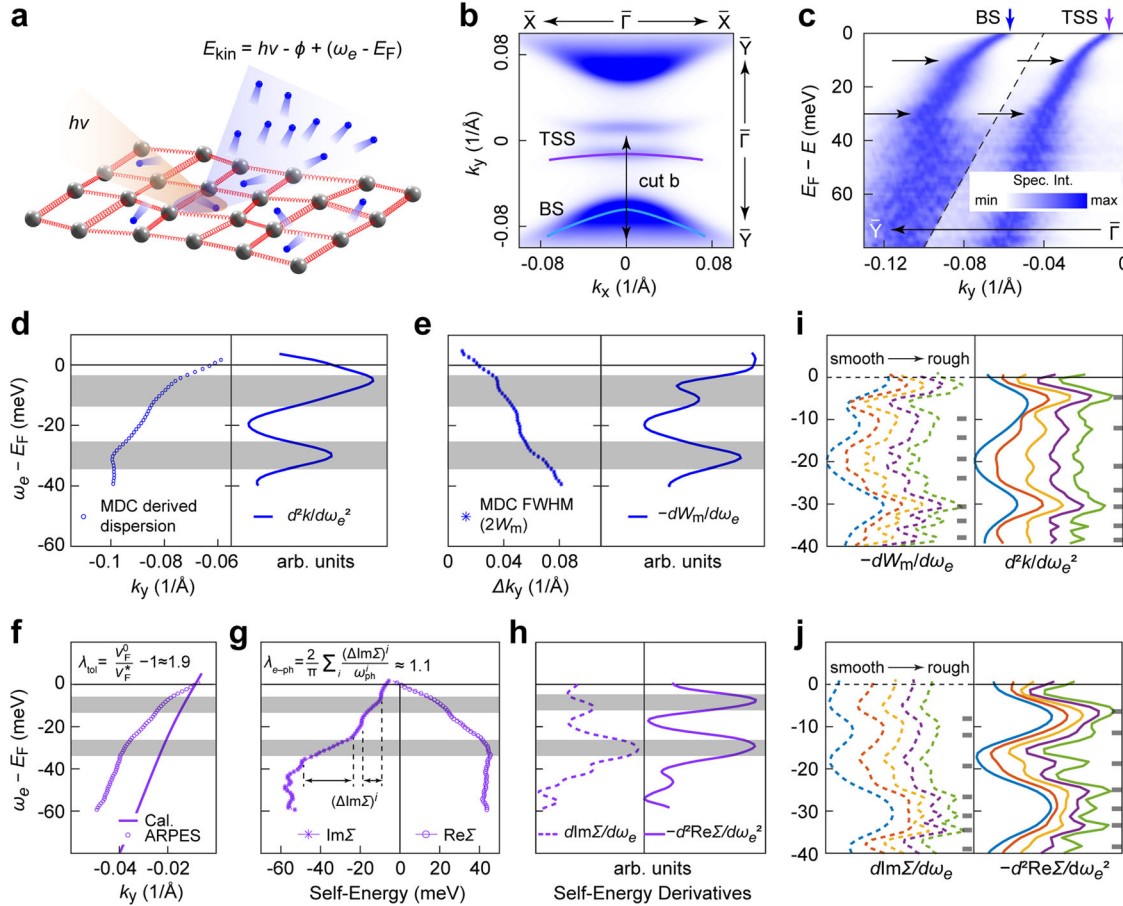

**Fig. 3 | Electronic band dispersion kinks observed by ARPES measurements. a** Schematic illustration of the photoemission process. Blue balls represent electrons. Gray balls connected with red springs represent the collective elastic arrangement of the lattice, phonons. $h\nu$ represents the incident photon energy. $E_{kin}$ represents the kinetic energy of the photoelectron. $\phi$ represents the work function. $\omega_e$ represents the electron initial energy. $E_F$ represents the Fermi energy. **b** Intensity plot of the Fermi surface mapping with the TSS and BS marked by purple and cyan curves, respectively. **c** ARPES measurements cutting along the $\bar{Y} - \bar{\Gamma}$ direction, as indicated by the double-headed arrow in (**b**). The presented data are normalized along the momentum-distribution-curve (MDC) direction in two separated momentum-energy areas. Four arrows indicate two dispersion anomalies (kinks) of both the TSS and the BS at binding energies of ~10 and ~30 meV. **d** Left panel: dispersion of the BS extracted by MDC fitting based on Lorentzian functions. Right panel: second-order derivatives of the dispersion $d^2k/d\omega_e^2$ showing peaks at the anomalies. **e** Left panel: full-width-at-half-maximum of MDCs ($2W_m$) extracted by

MDC fitting based on Lorentzian functions. Right panel: first-order derivatives of $W_m$ showing peaks at the anomalies. **f** Comparison between the MDC-derived band dispersion and corresponding first-principles calculated single-particle band dispersion. **g** The imaginary part (left panel) and real part (right panel) of the electron self-energy as functions of electron energies. The steplike increases of the imaginary part of the self-energy, $(\Delta Im\Sigma)^i$, are estimated to be $(\Delta Im\Sigma)^1 \approx 9.7$ meV and $(\Delta Im\Sigma)^2 \approx 24.9$ meV between adjacent plateaus. **h** First-order derivatives of the imaginary part of the self-energy $dIm\Sigma/d\omega_e$ (left panel) and second-order derivatives of the self-energy $-d^2Re\Sigma/d\omega_e^2$ as functions of electron energies showing peaks at anomalies. The kink energies are highlighted by the gray shaded areas in (**d–h**). **i** First-order derivatives of $W_m$ (left panel) and second-order derivatives of dispersion (right panel) of the BS based on different smooth curves that fit the experimental data. **j** Derivatives of the self-energy of the TSS (same as **h**) based on different smooth curves that fit the experimental data. The observed phonon modes are indicated by the gray lines in (**i**) and (**j**).

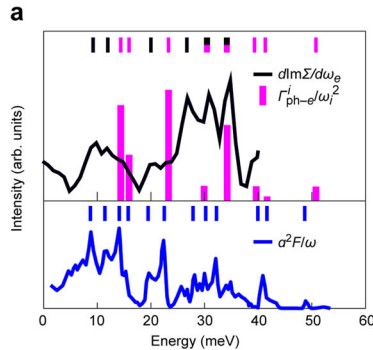

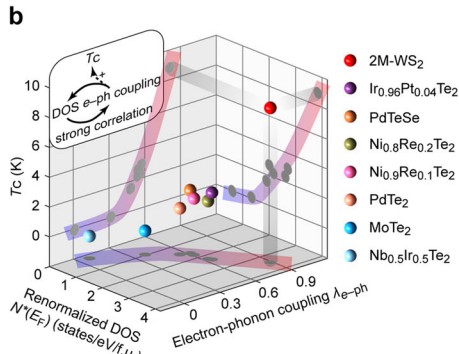

**Fig. 4 | Mode-selective EPC and superconductivity enhancement. a** Comparison among ARPES derived $d\mathrm{Im}\Sigma/d\omega_e$, Raman spectroscopy derived $\Gamma^i_{\mathrm{ph}-e}/\omega_i^2$, and first-principles calculated $\alpha^2F/\omega$ (extracted from ref. 22), where $\Sigma$ is the electron self-energy, $\omega_e$ is the electron energy, $\Gamma_{\mathrm{ph}-e}$ is the mode-selective magnitude of phonon linewidth due to phonon–electron scattering fitted based on Eq. (1), $\omega_i$ is the energy of the phonon mode $i$ experimentally extracted from the Raman spectra measured at the lowest temperature (5 K) by fitting to a Lorentzian function, $\alpha^2F$ is the Eliashberg function as a function of the phonon energy $\omega$. Phonon modes involved in EPC identified by ARPES, Raman spectroscopy, and first-principles calculation are indicated by black, magenta, and blue lines, respectively. **b** Compilation of superconducting transition temperature $T_C$ in topological T(T′, Td, and 2 M)-type TMD family as a function of renormalized DOS $N(E_F)$ and EPC strength $\lambda_{e-\mathrm{ph}}$. These parameters are extracted from temperature-dependent transport and specific heat measurements[22,27,42–47] and are summarized in Supplementary Table 3. Inset: electron–phonon ($e$–ph) coupling, the density of states (DOS), and electron correlations can enhance each other, and they synergistically enhance the superconducting transition temperature $T_C$.

parabolic imaginary part resulting from the electron correlations has been effectively excluded, the value of $\lambda_{e-\mathrm{ph}}$ is smaller than that of $\lambda^{\nu_F}_{\mathrm{tot.}} \approx 1.9$ derived from the renormalized Fermi velocity as mentioned above. Both kinks at around 10 and 30 meV are gradually weakened with elevating temperature (Supplementary Fig. 5), a phenomenon also noted in kinks of cuprate superconductors[37].

Notably, the self-energy analysis involving derivatives is sensitive to the smooth curves that fit the experimental data[38]. By adjusting the smoothing parameter, both the BS and the TSS exhibit finer structures, as presented in Fig. 3i, j, respectively. A double-peak feature emerges from the peak profile at approximately 10 meV, whereas four distinct peaks are distinguished from the peak profile at around 30 meV, accompanied by an additional less pronounced peak observed at around 20 meV. These results reveal reliable phonon modes coupling in the electron self-energy rather than experimental artifacts since they are ubiquitous for the BS and the TSS derived from both real and imaginary parts of self-energies.

Theoretically, the EPC characters are fully described by the Eliashberg function $\alpha^2F(\omega)$[39], indicating the total transition probability of a quasiparticle by coupling with a phonon mode of energy $\omega$[40,41]. The accumulated EPC strength $\lambda_{e-\mathrm{ph}}$ (also referred to as mass enhancement factor) is integrated from the Eliashberg function as ref. 39

$$\lambda_{e-\mathrm{ph}} = 2\int_0^{\omega_{\max}} \frac{\alpha^2F(\omega)}{\omega} d\omega. \tag{4}$$

Experimentally, the EPC can be extracted from temperature-dependent Raman phonon linewidth [see Eq. (2)][29,30] or identified from the steplike increase of the imaginary part of the electron self-energy [see Eq. (3)][36]. We make a comparison between the experimental results ($d\mathrm{Im}\Sigma/dE$ and $\Gamma^i_{\mathrm{ph}-e}/\omega_i^2$) with the theoretical calculation ($\alpha^2F/\omega$) in Fig. 4a, showing nice consistency. The two modes with the lowest energies (~10 meV) observed by ARPES are attributed to acoustic modes. Phonon modes with zero momentum (e.g., $\omega \sim 30$, 35 meV, $q = \Gamma$) are observed in both ARPES and Raman spectroscopy characterizations. The phonon modes with non-zero momenta (e.g., $\omega \sim 20$ meV, $q = N$) are observable in ARPES only.

## Discussion

With the systematic study on the EPC of 2M-WS$_2$, we propose that the renormalized DOS and strong EPC could reinforce each other and

synergistically enhance the $T_C$ of 2M-WS$_2$, similar to a recent understanding of cuprates[11]. For a clearer view, we compile the parameters of EPC-induced superconductivity in the topological T(T′, Td, and 2M)-type TMD family that share similar band structures[22,27,42–47] such as band inversion near $\bar{\Gamma}$, and the TSS guaranteed by the trigonal crystal field[48] (Fig. 4b and Supplementary Table 3). It is evident that $T_C$ is positively correlated with both the EPC strength $\lambda_{e-\mathrm{ph}}$ and renormalized DOS. The highest $T_C$ (8.8 K) of 2M-WS$_2$ coincides with the strongest EPC strength $\lambda_{e-\mathrm{ph}}$ (0.79)[22] and the largest renormalized DOS $N(E_F)$ (3.81 states/eV/f.u.)[27]. Similar ARPES and Raman spectroscopy characterizations have been performed on the reference material 2M-WSe$_2$ showing no experimental signature of strong EPC (see Supplementary Figs. 8 and 9). We thus suggest that the difference in EPC strengths between 2M-WS$_2$ and 2M-WSe$_2$ leads to their distinct superconducting behaviors[49].

Our discovery of strong and mode-selective EPC in 2M-WS$_2$ is essential for comprehending the formation of topological superconductivity and offers valuable perspectives into other exotic properties that are currently not fully understood. We have reproduced the unusual temperature and magnetic field dependence of transport behaviors, which were previously reported as indications of crossover between a Fermi liquid and a strange metal state[21]. We propose that many-body interactions, including both electron–electron correlations and EPC, should be considered, as this observation could be alternatively interpreted as a phonon-drag dominated transport region below ~ 80 K due to strong EPC (Supplementary Fig. 6). The mysterious striped surface charge order that coexists with superconductivity and suppresses Majorana bound states[20] could potentially originate from electronic states coupled with acoustic phonons, which is indicated by the coincident charge order wave vector observed by STM measurements and the wave vector of the acoustic phonon mode that contributes to the peak of Eliashberg function[22] (Supplementary Fig. 7). The multiple saddle-like band structures near $E_F$ might also play a role in the formation of the charge orders, as it is proposed in 2H-TaSe$_2$ that the charge density orders can be effectively tuned by a van Hove singularity[50].

We also notice that the temperature dependence of transport properties in 2M-WS$_2$ shows similarity to that in high-$T_C$ cuprates[51–53]. The origin of such behavior is debatable, with various theories proposed, including the "two-scattering-time" model[54,55], the 1D stripe transport model[53], as well as phonon drag and multiband conduction[56]. Furthermore, both 2M-WS$_2$ and high-$T_C$ cuprates show

pronounced low-energy kinks in the electronic dispersions. These intriguing similarities indicate that 2M-WS$_2$ might become an important reference compound in the study of unconventional high-$T_C$ superconductors.

## Methods

### Sample synthesis

2M-WS$_2$ single crystals were prepared by the deintercalation of inter-layer potassium cations from K$_{0.7}$WS$_2$ crystals. For the synthesis of K$_{0.7}$WS$_2$, K$_2$S$_2$ (prepared via liquid ammonia), W (99.9% Alfa Aesar), and S (99.9%, Alfa Aesar) were mixed by the stoichiometric ratios and ground in an argon-filled glovebox. The mixtures were pressed into a pellet and sealed in the evacuated quarts tube. The tube was heated at 850 °C for 2000 min and slowly cooled to 550 °C at a rate of 0.1 °C min$^{-1}$. The synthesized K$_{0.7}$WS$_2$ (0.1 g) was oxidized chemically by K$_2$Cr$_2$O$_7$ (0.01 mol L$^{-1}$) in aqueous H$_2$SO$_4$ (50 ml, 0.02 mol L$^{-1}$) at room temperature for 1 h. Finally, the 2M-WS$_2$ crystals were obtained after washing in distilled water several times and drying in the vacuum oven at temperature[14].

### ARPES measurements

High-resolution laser-based angle-resolved photoemission spectroscopy (laser-ARPES) measurements were performed at a home-built setup ($h\nu = 6.994$ eV) at ShanghaiTech University. The sample was cleaved in situ and aligned the $\bar{\Gamma} - \bar{Y}$ direction parallel to the analyzer slit. The measurements were carried out under ultra-high vacuum below $5 \times 10^{-11}$ Torr. The measurement temperature is about 20 K (above $T_C = 8.8$ K). Data were collected by a DA30L analyzer. The total energy and angle resolutions were ~1 meV and ~0.2°, respectively.

### Electronic band structure calculations

The density functional theory (DFT) calculations were carried out via the *Vienna Ab initio Simulation Package* (VASP)[57]. The projector-augmented wave (PAW) method and the plane-wave basis with an energy cutoff of 400 eV were adopted. The exchange-correlation energy was approximated by the Perdew-Burke-Ernzerhof (PBE) type generalized gradient approximation (GGA)[58]. The experimental lattice constants (the cif documents) were taken from ref. 14. The structural relaxation was performed to optimize lattice constants and atomic positions, with a force criterion of 0.01 eV/Å, and the DFT-D3 method[59] was used to include van der Waals corrections. Spin-orbit coupling was included in self-consistent calculations. The convergence thresholds were $10^{-8}$ eV and $10^{-5}$ eV/Å for energy and force, respectively. Topological properties, including surface state calculations, were performed with WannierTools package[60], based on the tight-binding Hamiltonians constructed from maximally localized Wannier functions (MLWFs) by the Wannier90 package[61–63].

### Phonon calculations

Phonons at the Γ-point were computed following the approach proposed by Porezag and Pederson[64]. Moreover, PHONOPY[65] codes were used for data postprocessing. In terms of phonon dispersion calculations, structural relaxations were performed with thresholds of $1.0 \times 10^{-4}$ eV/Å and $1.0 \times 10^{-8}$ eV for higher accuracy, and then dispersion relations were calculated by VASP[57]. The force constants were calculated using density functional perturbation theory (DFPT)[57] in a $3 \times 3 \times 3$ supercell with a $9 \times 9 \times 9$ k-mesh by VASP[57]. The phonon dispersion was then obtained using PHONOPY[65]. To compare with Raman experiments, we identified the phonon modes with Raman activity ($A_g$ and $B_g$ modes) based on the crystalline space group and marked them in the calculated Fig. 1e.

### Raman spectroscopy

Raman scattering measurements were conducted in the back-scattering geometry with an excitation wavelength of 532 nm and an incident power of 0.6 mW. The laser beam was focused to a spot size of ~1 μm using a 40 × objective. Bragg notch filters were used to filter the laser line. The scattered signal was dispersed by an 1800 grooves/mm grating and detected by a liquid-nitrogen-cooled charge-coupled device. Temperature control was achieved using a Montana Instruments Cryostation. Polarization-angle dependent Raman scattering was performed using a half-wave plate mounted on a rotation stage. Two polarizers placed upstream and downstream of the sample chamber in the optical path were used to control the polarization configuration of the incident and scattered beams. The circularly polarized measurements were performed using the same optical layout by replacing the half-wave plate with a quarter-wave plate, as detailed in ref. 66.

The Raman tensors for the phonon modes can be described as:

$$\hat{R}_{A_g} = \begin{pmatrix} a & 0 & d \\ 0 & b & 0 \\ d & 0 & c \end{pmatrix}, \hat{R}_{B_g} = \begin{pmatrix} 0 & e & 0 \\ e & 0 & f \\ 0 & f & 0 \end{pmatrix}. \tag{5}$$

Using these Raman tensors, we can calculate the Raman intensity in the parallel (XX) and perpendicular (XY) polarization in the back-scattering configuration used in our experiments. The polarization-angle dependence of the mode intensity can be calculated as:

$$S_{A_g}^{\parallel} \propto \left( a\cos^2\theta + b\sin^2\theta \right)^2, S_{A_g}^{\perp} \propto (b-a)^2 \sin^2(2\theta),$$
$$S_{B_g}^{\parallel} \propto e^2 \sin^2(2\theta), S_{B_g}^{\perp} \propto e^2 \cos^2(2\theta). \tag{6}$$

Here, the parallel and perpendicular symbols denote the XX and XY polarizations, respectively. For the details of the polarization-angle dependent Raman spectroscopy, please refer to Supplementary Fig. 1.

## Data availability

All data are processed by MATLAB. The authors declare that the data supporting the findings of this study are available within this article and the Supplementary Information file. All raw data generated during the current study are also available from the corresponding author upon request.

## Code availability

Results can be reproduced using standard VASP packages. Methods are fully described. Codes used to produce figures can be made available upon request.

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

## Acknowledgements

Y.L. acknowledges support from the National Natural Science Foundation of China (NSFC) (Grant No. 12104304), Hubei Provincial Natural Science Foundation of China (Grant No. 2024AFB935), and the Fundamental Research Funds for the Central Universities (2042023kf0107). N.X. acknowledges support from the National Natural Science Foundation of China (NSFC) (Grant No. 12274329). X.X. acknowledges support from the Natural Science Foundation of Jiangsu Province (Grant No. BK20231529). Y.F. acknowledges support from the Shanghai Rising-Star Program (23QA1410700). Z.L. acknowledges the support from the National Nature Science Foundation of China (NSFC) (Grants No. 92365204, 12274298) and the National Key R&D Program of China (Grant No. 2022YFA1604400/03).

## Author contributions

Y.L. and L.X. conceived the project. Y.L., L.X., and H.Z. carried out ARPES measurements with the assistance of N.X., Z.L., and Y.C.; Y.L., L.X., S.D., E.L., and G.Z. performed the data analysis on the ARPES results. G.L. carried out Raman spectroscopy measurements. G.L., X.X., and Y.L. performed the data analysis on the Raman spectroscopy results. Y.F. and F.H. synthesized the single crystals. L.X. performed first-principles calculations of electronic structures and phonon dispersions. Y.L. and L.X. wrote the first draft of the paper. X.X., S.Z., S.L., L.Y., Z.L., and N.X. contributed to the revision of the manuscript. All authors contributed to the scientific planning and discussions.

## Competing interests

The authors declare no competing interests.
