## [Peer Review File · Nature Communications]

Evidence of strong and mode-selective electron–phonon coupling in the topological superconductor candidate 2M-WS₂Editorial Note: Parts of this Peer Review File have been redacted as indicated to remove third-party material where no permission to publish could be obtained.

Editorial Note: Authors have permission to reproduce the figure panels from other NCOMMS publications.

REVIEWER COMMENTS

Reviewer #1 (Remarks to the Author):

The authors reported Raman scattering measurements showing nonmonotonic phonon linewidths with temperature, along with ARPES results showing dispersion kinks at different binding energies, to explore the renormalized properties in an electron-phonon superconductor 2M-WS₂. By fitting to a phenomenological linewidth model, the linewidth from phonon-electron scattering for each Raman mode was deduced and used to estimate the coupling strength of these phonon modes. Moreover, the possible phonon features deduced from ARPES experiments were compared with the theoretical Eliashberg function. The present work is of interest for understanding the electron-phonon coupling in 2M-WS₂, but there remain many questions that need to be addressed before publication, as described below:

(1) The calculated Fermi level DOS (1.38 states/eV/f.u.) in Fig. 1g is argued to be much smaller than 3.81 states/eV/f.u. derived from a Sommerfeld coefficient estimated from specific heat data [PRB 102, 024523 (2020)], however, I cannot find the latter DOS value when going through this PRB work. Also, it is known that the Sommerfeld coefficient at low temperature is proportional to $N_F(1+\lambda)$, but given the present experimental work not accurately measuring the coupling strength (λ), it should be improper to speculate the Fermi level DOS from the Sommerfeld coefficient and argue about many-particle correlations.

(2) The fitted linewidths from phonon-electron scattering for the measured Raman modes are questionable. First, not all fitting parameters are given, and it is found from Fig.2/ Fig. S2 that the linewidth from phonon-phonon scattering is constant for many modes, deviating from the expected trend of monotonic increase with temperature. Second, based on equation 2, the coupling strength for all Raman modes should be $0.32 \times 3.81 / 1.38 = 0.88$ when adopting the theoretical Fermi level DOS as mentioned above. Then it is strange to see this value, not including contributions from non-Raman modes at Gamma and many other non-zone-center phonon modes, to be already larger than the total coupling strength of 0.8 for 2M-WS₂.

(3) It is not clear whether the phonon dispersions in Fig. 1e are extracted from other works. The authors wrote in the method section they calculated the phonons at the Gamma point, but no method and calculation details about the phonon dispersions were presented.

(4) In line 187, the authors refer to a total coupling strength of 1.9 from the renormalized Fermi velocity for energy bands along a particular direction, whereas 2M-WS₂ exhibits a 3D Fermi surface and energy bands along other different directions need to be probed.

(5) Another issue is about comparison with the Eliashberg function in Fig. 4(a), which involves contributions not only from the coupling strength of each phonon mode but also from the phonon density of states. Thus is it reasonable to directly compare the peak positions of $\alpha_2 F(\omega)$ to those derived from Raman or ARPES experiments?

Reviewer #2 (Remarks to the Author):

The authors investigated the electron-phonon coupling (EPC) in the 2M stacking topological superconductor WS₂ by performing Raman scattering and angle-resolved photoemission spectroscopy (ARPES). They proposed that the strong EPC and thus the large renormalized DOS could account for the highest superconducting transition temperature T_c observed in 2M-WS₂ (8.8 K) among all transition metal dichalcogenides. They tried further to explain the exotic properties like strange metal behavior in transport properties by using the phonon drag effect.

The paper provided two main indications in support of the strong EPC in this system. One is the nonmonotonic temperature dependence of the phonon linewidths; the other is the appearance of kinks in the dispersion of the low-energy electronic state measured by ARPES. However, both pieces of evidence have their own weakness.

1. The interpretation of the Raman peak line width heavily depends on the model in eq. (1). But is that model the only way to interpret the data? The authors did not discuss the physical insight of the extra terms in eq. (1), corresponding to electron-phonon coupling. How reliable is this attribution?
2. The kinks in the dispersion curves in Fig. 3 and Fig. S3 appeared very poorly resolved. It is not clear whether argument made upon this is reliable.
3. As the authors found, the strongly renormalized electronic density of states indicate the presence of correlated many-body physics. They also proposed that both the renormalized DOS and strong EPC are the key ingredients for the relatively higher T_c in 2M-WS₂. While it is understandable that the origin of the strong correlations is beyond the scope of the paper, it is certainly not EPC alone. Therefore, when the authors commented on the two recent papers (Refs. 20 and 21) using the electron-phonon coupling picture alone, the argument is clearly stretched.
4. In Fig. 6 of Supplementary Material, a phonon drag model is used to fit the temperature dependence of resistivity and there is also a phonon drag dominated region in Hall effect. Such a fitting may not be reliable. In Fig.6(b) the model fits the data well up to $T \sim 60$ K, while in Fig.6(d), the model deviates from the data as $T > 40$ K. However, both fits produce similar T_0 value. In Fig. (a) and (c), I do not know how the phonon dominated region can be determined reliably. How is $T_{\text{crossover}}$ determined in Fig.6 (c)? Moreover, in the phonon drag model, how is the magnetic field dependence of Hall coefficient (R_H) obtained?
5. The temperature dependence of R_H in 2M-WS₂ is very similar to that in high- T_c cuprates (especially in slightly underdoped region). In cuprates, the origin of such a behavior is still under debate, and many works believe it is associated with some exotic scenarios, for example, the two different relaxation times related to different electron degrees in a strongly correlated system. Can a simple phonon drag model explain the Hall effect in both cuprates and 2M-WS₂?

Reviewer #3 (Remarks to the Author):

Please refer to the attached file.

In this paper, the authors have quantitatively evaluated the strength of electron-phonon coupling in the topological superconductor 2M-WS₂, which has a relatively high superconducting transition temperature T_c (~ 8.8 K), by complementary use of Raman scattering and ARPES to investigate the origin of its high T_c . The results are consistent with the previous study based on the Migdal-Eliashberg theory, and the authors have concluded that it can be explained by the large density of states at $N(E_F)$ and the strong electron-phonon coupling from the comparison with other transition-metal dichalcogenides.

It is a very interesting result, and the paper basically seems to be worth being published in a high impact journal such as *Nature Communications*.

However, I have several concerns that I would like the authors to address before accepting the paper, and if possible, improve the manuscript.

First, I would like to see a little more detail on the fitting using Eq. (1) in line 133.

In the caption of Supplementary Fig. 2, it says “Upper panel: Fitted temperature-dependent phonon energy of 8 observed Raman-active phonon modes”. What kind of the fitting was performed here? The fitting to the Lorentzian? This should be clarified for the caption of upper panels.

In line 131, it says “the observed nonmonotonic temperature dependent linewidths can be perfectly fitted by the following formula”, but how was ω_e in Eq. (1) treated in the fitting process? Treated as a fitting parameter? Wouldn't it be more reasonable to multiply $N(\omega_e)$ or $N(\omega_e + \omega_o)$ and integrate? For ω_o , was the phonon energy shown in the upper panels of Supplementary Fig. 2 used for each mode?

In line 147, it says $\lambda_{e-ph}^{Raman} = 0.32$, but according to Eq (2), because both the phonon line width Γ_i and phonon energy ω_i seems to show temperature dependence, it is assumed that λ_{e-ph}^{Raman} also shows temperature dependence. Were the values extracted from the Raman measurements at the lowest temperature 5 K used? Or for the phonon line width Γ_i , was Γ_{ph-e} obtained by the fitting using Eq (1) used for each mode?

In Table I, ω_0^{Raman} is stated as the value extracted from the Raman measurements at the lowest measurement temperature 5K, but is the linewidth Γ_{ph-e}^i in Table I not the value extracted from the Lorentzian fitting of the Raman profile at 5K? Or is it a value of Γ_{ph-e} obtained by fitting using Eq (1) for each mode?

The authors should clarify how the values of Γ_i and ω_i were obtained to deduce $\lambda_{e-ph}^{Raman} = 0.32$ using Eq. (2).

In Eq. (3) of line 184, it says $\lambda_{e-ph} \approx 1.1$, but again, it has not been clearly stated how the values used for ω_{ph}^i and $(\Delta\text{Im}\Sigma)^i$ were obtained. Was ω_0^{Raman} used for ω_{ph}^i ? For $(\Delta\text{Im}\Sigma)^i$, it seems to have been shown in Fig. 3g how to evaluate $(\Delta\text{Im}\Sigma)^i$, but it should be mentioned in the figure caption and main text.

In line 174, it says $\lambda_{tot}^{DOS} = 1.9$, but according to $N_0 = 1.38$ in line 105 and $N(E_F) = 3.81$ in line 107, $N(E_F)/N_0 - 1$ should become 1.76. I think it may be reasonably consistent, but "matches perfectly" seems to be an overstatement.

Finally, while Fig. 4b is a very interesting plot, I have a concern that it does not include 2M-WSe₂, which seems to be the most suitable reference material. According to the co-authors of this paper, 2M-WSe₂ does not seem to exhibit superconductivity at ambient pressure, but it does under high pressure, and the ARPES measurements have already been performed; is there any valid reason not to perform similar measurements and analyses for 2M-WSe₂ for direct comparison? If there is no valid reason and it is feasible to perform them, the results should be included to validate the authors' conclusions.

If the authors can appropriately answer to these comments, I would then like to recommend this paper for publication in *Nature Communications*.

Reply to Reviewer #1:

Reviewer #1: The authors reported Raman scattering measurements showing nonmonotonic phonon linewidths with temperature, along with ARPES results showing dispersion kinks at different binding energies, to explore the renormalized properties in an electron-phonon superconductor 2M-WS₂. By fitting to a phenomenological linewidth model, the linewidth from phonon-electron scattering for each Raman mode was deduced and used to estimate the coupling strength of these phonon modes. Moreover, the possible phonon features deduced from ARPES experiments were compared with the theoretical Eliashberg function. The present work is of interest for understanding the electron-phonon coupling in 2M-WS₂, but there remain many questions that need to be addressed before publication, as described below:

Authors' response:

We thank the Referee for his/her pertinent reviewing and greatly appreciate the Reviewer for acknowledging that “the present work is of interest for understanding the electron-phonon coupling in 2M-WS₂”. We also thank the Reviewer for his/her detailed comments and suggestions and we will provide a point-to-point response addressed to the reviewer as follows.

(1) The calculated Fermi level DOS (1.38 states/eV/f.u.) in Fig. 1g is argued to be much smaller than 3.81 states/eV/f.u. derived from a Sommerfeld coefficient estimated from specific heat data [PRB 102, 024523 (2020)], however, I cannot find the latter DOS value when going through this PRB work. Also, it is known that the Sommerfeld coefficient at low temperature is proportional to $N_F(1+\lambda)$, but given the present experimental work not accurately measuring the coupling strength (λ), it should be improper to speculate the Fermi level DOS from the Sommerfeld coefficient and argue about many-particle correlations.

Authors' response:

We thank the Referee for his/her pertinent comments.

We would kindly like to begin by elucidating these two concepts used in the manuscript, the noninteracting single-particle DOS at E_F $N_0(E_F)$ and the renormalized (interacting many-particle) DOS at E_F $N^*(E_F)$. The noninteracting DOS $N_0(E_F)$, namely, is derived from the (calculated) band structure that excludes any correlation effects, such as electron-phonon coupling and electron-electron correlations. Since many-particle correlations are more or less involved in any material systems, $N_0(E_F)$ is a theoretical concept that can be obtained from *ab initio* calculation. On the other hand, the renormalized (interacting) DOS $N^*(E_F)$ contains the total quasiparticle contributions of the interacting system and can possibly be obtained from experiments. $N^*(E_F)$ and $N_0(E_F)$ are linked by $N^*(E_F) = N_0(E_F) \times (1 + \lambda_{\text{tot}})$, where λ_{tot} denotes the total many-particle correlation strength (not only electron-phonon coupling) and is termed as (total) coupling constant, mass enhancement factor or renormalization factor in the literature. Therefore, one can estimate the correlations from the ratio of $N^*(E_F)$ and $N_0(E_F)$. Similar analysis approaches are commonly used in literature [e.g., *Phys. Rev. B* 83, 012502 (2011) and *Phys. Rev. B* 52, 16165 (1995)].

In our manuscript, $N_0(E_F) = 1.38$ states/eV/f.u. is obtained from our noninteracting band structure calculation, whereas $N^*(E_F)$ is proportional to the Sommerfeld coefficient $\gamma = 8.97 \pm 0.08$ mJ/mol/K² [*Phys. Rev. B* 102, 024523 (2020)] as $N^*(E_F) = \frac{3\gamma}{\pi^2 k_B^2} = 3.81 \pm 0.03$ states/eV/f.u. (where k_B is the Boltzmann constant, please also refer to Supplementary Table 2). Hence, as argued above, the many-particle renormalization factor is estimated as $\lambda_{\text{tot}}^{\text{DOS}} = \frac{N^*(E_F)}{N_0(E_F)} - 1 = 1.76$. Noticeably, the comparison between ARPES and the noninteracting single-particle band structure calculation leads to a nicely consistent estimation of $\lambda_{\text{tot}}^{\text{VF}} = \frac{v_F^0}{v_F^*} - 1 \approx 1.9$, where v_F^0 and v_F^* are calculated single-particle “bare” Fermi velocity and ARPES measured many-particle renormalized Fermi velocity, respectively. Therefore, both specific heat measurements (the Sommerfeld coefficient) and ARPES results indicate strong correlated many-particle physics in 2M-WS₂.

In the revised manuscript, we have clarified the definition of $N_0(E_F)$ and $N^*(E_F)$ accordingly to eliminate the confusion and included the derivation of $N^*(E_F)$ from the Sommerfeld coefficient.

(2) The fitted linewidths from phonon-electron scattering for the measured Raman modes are questionable. First, not all fitting parameters are given, and it is found from Fig.2/Fig. S2 that the linewidth from phonon-phonon scattering is constant for many modes, deviating from the expected trend of monotonic increase with temperature. Second, based on equation 2, the coupling strength for all Raman modes should be $0.32*3.81/1.38 = 0.88$ when adopting the theoretical Fermi level DOS as mentioned above. Then it is strange to see this value, not including contributions from non-Raman modes at Gamma and many other non-zone-center phonon modes, to be already larger than the total coupling strength of 0.8 for 2M-WS₂.

Authors' response:

We thank the Reviewer for his/her insightful questions.

For the first question, the temperature dependence of the Raman linewidths is fitted based on equation 1 including 4 fitting parameters (Γ_0 , Γ_{ph-ph} , Γ_{ph-e} , and ω_e , please note that ω_0 is the optical phonon energy at zero temperature and we set it to the Raman measured frequency at the lowest temperature $\omega_{T=5K}$). Following the Reviewer's suggestion, we have presented the complete results of fitting parameters in the revised Supplementary Material.

As shown in the second term of equation 1, the linewidth from phonon-phonon scattering $\Gamma_{ph-ph} \times \frac{2}{\exp\left(\frac{\omega_0}{2k_B T}\right)+1}$ guarantees its monotonic increase with temperature and it converges to zero at zero temperature. Considering that the temperature invariant term Γ_0 (due to trivial impurity scattering, boundary scattering, etc.) can be significantly larger than phonon-phonon scattering term and phonon-electron scattering term for some modes [e.g., $A_g(2)$ as shown in Fig. 2f], the fitted results of these two scattering mechanisms presented in Fig. 2/Fig. S2 are upshifted by a constant background Γ_0 for a better illustration (as indicated in the caption). The Reviewer's impression that "the linewidth from phonon-

phonon scattering is constant for many modes” is probably due to a relatively small contribution from phonon-phonon scattering compared to the large temperature invariant background Γ_0 . To avoid this misunderstanding, we have marked Γ_0 by black arrows in the revised Fig. 2/ Fig. S2, as also shown below.

Fig. R1. The lower panel of revised Fig. 2e-g.

The second question is insightful and enlightening. By careful literature research, equation 2 in our manuscript can be traced back to the pioneer electron-phonon coupling study by Phillip B. Allen [equation 12 in *Phys. Rev. B* 6, 2577 (1972)]. The equation in the literature is written as

$$\sum_Q \gamma_Q / \omega_Q^2 = \frac{1}{4} \pi N(0) \lambda,$$

where γ_Q is the phonon linewidth which has the same definition as Γ_i in our manuscript (half-width-at-half-maximum of the Lorentzian profile, please refer to equation 8 in this literature), ω_Q is the phonon frequency, $N(0)$ is the **noninteracting** electronic density of states (DOS) at the Fermi surface for both spin orientations per unit cell, and λ is the electron-phonon coupling constant. In our manuscript, the correct form of Eq. 2 should have an additional factor 4 (which has been omitted in the original manuscript):

$$\lambda_{e-ph} = \sum_i \frac{4\Gamma_i}{\pi N_0(E_F) \omega_i^2}.$$

Surprisingly, the coupling strength for all Raman modes is $0.32 \times 4 \times 3.81 / 1.38 = 3.53$, which is well above the calculated electro-phonon coupling constant $\lambda_{e-ph}^{cal.} = 0.79$.

We argue that the breakdown of Eq. 2 is possibly due to strong electron correlation in 2M-WS₂ which is beyond the framework of Phillip B. Allen’s theoretical work. The strong

electron correlation in 2M-WS₂ is supported by a relatively large total coupling constant ($\lambda_{tot.} \sim 1.9$), well exceeding the electron-phonon coupling constant ($\lambda_{e-ph} = 0.79$). It would be natural to expect much larger “noninteracting” $N_0(E_F)$ to calculate electron-phonon coupling strength of a correlated system. However, the knowledge about electron-phonon coupling of strongly correlated systems, such as 2M-WS₂ and other high-temperature superconductors, requires more sophisticated theoretical models [*Physics Reports* 338, 1-2, 1-264, (2000)], which is beyond the scope of our manuscript.

In the revised manuscript, the form of equation 2 has been corrected and a discussion on its breakdown (as mentioned above) is included.

(3) It is not clear whether the phonon dispersions in Fig. 1e are extracted from other works. The authors wrote in the method section they calculated the phonons at the Gamma point, but no method and calculation details about the phonon dispersions were presented.

Authors’ response:

We thank the Referee for pointing this out. We have independently performed the phonon dispersion calculations in Fig. 1e, which is in general agreement with a previous study [*Nano Lett.* 21, 1, 709-715 (2021)]. In the revised manuscript, methods and calculation details on the phonon dispersions are included in the Methods section.

(4) In line 187, the authors refer to a total coupling strength of 1.9 from the renormalized Fermi velocity for energy bands along a particular direction, whereas 2M-WS₂ exhibits a 3D Fermi surface and energy bands along other different directions need to be probed.

Authors’ response:

We thank the Referee for this constructive comment.

As mentioned by the Referee, we quantitatively estimate the total coupling strength of 1.9 from the renormalized Fermi velocity for a specific band along the $\bar{\Gamma} - \bar{Y}$ direction. We fully agree with the Referee that 2M-WS₂ indeed exhibits an anisotropic 3D Fermi surface

and strictly speaking, to obtain the total coupling strength, it is necessary to measure the Fermi velocity renormalization at all Fermi momenta of the whole Fermi surface.

In the (revised) manuscript, we have clarified the distinction between the coupling strength for a specific band and the total coupling strength averaged over the whole 3D Fermi surface. We also report the nice coincidence of these two quantities extracted from the renormalized Fermi velocity by ARPES measurements and the renormalized DOS by temperature-dependent specific heat measurement. This coincidence implies the momentum dependence of the many-body coupling in 2M-WS₂ might be weak. The suggested weak momentum dependence is also supported by the isotropic (momentum-invariant) superconducting gap [*Nat. Commun.* 12, 2874 (2021)], in contrast to the momentum-dependent electron-phonon coupling and the superconducting gap in 2H-NbSe₂ [*Phys. Rev. Lett.* 92, 086401 (2004)].

Technically, it is extremely challenging to measure the Fermi velocity renormalization for the whole 3D Fermi surface. A complete 3D Fermi surface mapping generally requires photon-energy-dependent synchrotron-based ARPES measurements. To reduce k_z broadening, high photon energies with long photoelectron mean-free-paths near E_F are favored. However, the energy and momentum resolution that is of paramount importance for the measurement of the renormalized Fermi velocity would be significantly reduced compared to the low-energy laser-ARPES. For the case of 2M-WS₂, the fine low-energy kink features are observable in the laser-ARPES measurements with high energy-momentum resolution but absent in the synchrotron-based ARPES measurements.

We have made extensive efforts to measure the band renormalization on the 2D (k_x - k_y) plane using laser-ARPES, as presented in Supplementary Fig. 4 and Fig. R2. The kink features can be observed along different momentum directions (Supplementary Fig. 4). The renormalized Fermi velocities are quantitatively extracted along 5 different momentum directions, as indicated by the magenta lines in Fig. R2a,b. Correspondingly, the “bare” Fermi velocities are extracted from the (non-interacting) *ab initio* calculation, as indicated by the red lines in Fig. R2a,b. Surprisingly, this analysis approach leads to a clear variance

of the coupling strength (1.9~9.3) for different bands and momentum directions, as shown in Fig. R2d.

Fig. R2. Fermi velocity renormalization along different momentum directions. **a**, ARPES measurements cutting with different k_x momenta. **b**, Corresponding non-interacting *ab initio* calculation. The magenta lines and red lines are linear fitting results of the local intensity maxima of **a** and **b** in the energy window between E_F and $E_F - 5$ meV. The slopes of these lines are Fermi velocities. **c**, Momentum-distribution-curves at $E_F - 5$ meV extracted from **a**. **d**, The fitting results.

However, we believe the variance is **not** intrinsic and does **not** evidence momentum dependence of the many-body interaction for the following reasons. The above analysis approach can be inaccurate considering the mixture of the surface and bulk states. The band β is a topological surface state along $k_x = 0$, which is separated with the bulk continuum in the k - E space [also see *Nat. Commun.* 12, 2874 (2021)]. Hence, it results in a reliable coupling strength (1.9). For momentum directions away from $\bar{\Gamma} - \bar{Y}$ ($k_x \neq 0$), the band β gradually merges into the bulk continuum. Similarly, the band α has increased bulk contribution for larger k_x . The bulk origins are experimentally evidenced by the momentum-distribution-curves (MDCs) with enhanced peak widths and asymmetric profiles, as shown

in Fig. R2c. The uncertainty of bulk dispersions due to k_z broadening can lead to unreliable results and significant overestimation of the coupling strength.

Based on the consideration above, we only performed the quantitative analysis for the band β along the $\bar{\Gamma} - \bar{Y}$ direction ($k_x = 0$). The momentum dependence of the many-body interaction is important for the understanding of the superconducting mechanism and can be relevant in 2M-WS₂, however, it requires further investigations.

(5) Another issue is about comparison with the Eliashberg function in Fig. 4(a), which involves contributions not only from the coupling strength of each phonon mode but also from the phonon density of states. Thus is it reasonable to directly compare the peak positions of $\alpha^2F(\omega)$ to those derived from Raman or ARPES experiments?

Authors' response:

We thank the Referee for his/her insightful comments.

We fully agree with the Referee that the Eliashberg function involves contributions from all phonon momenta q . For Raman measurements, only the Raman-active modes near Γ ($q \approx 0$) contribute. Therefore, we didn't expect to capture the complete features of the Eliashberg function but a fraction of it by a comparison with the Raman phonon linewidths. This is reasonable since all Raman-active modes do contribute to corresponding peak features of both phonon density of states $F(\omega)$ and the Eliashberg function $\alpha^2F(\omega)$, as shown in Fig. 1c of *Nano Lett.* 21, 1, 709 (2021).

ARPES measurements, on the other hand, are good supplements to the Raman measurements. The low-energy electronic self-energy of a specific band is contributed by coupling between the electronic states and all phonon modes. As presented in Fig. 4a, the peaks of $d\text{Im}\Sigma/d\omega_e$ correspond to phonon modes with both zero and nonzero momenta q . It has been widely adopted in the study of high- T_c superconductors that the Eliashberg function can be deduced from the first derivation of the imaginary part of the self-energy [please refer to Fig. 9 of *Phys. Rev. B* 8, 174516 (2010)] or the second derivation of the real part of the self-energy [also see references: *Phys. Rev. Lett.* 95, 117001 (2005) & *Phys. Rev. B* 8, 174516

(2010)]. Therefore, we believe the approach employed in this paper is feasible and reasonable.

In summary, we are truly thankful for the first Referee's helpful comments and suggestions, which have significantly helped us in further exploring physics in this material and elucidating the experimental details.

Reply to Reviewer #2:

Reviewer #2: The authors investigated the electron-phonon coupling (EPC) in the 2M stacking topological superconductor WS₂ by performing Raman scattering and angle-resolved photoemission spectroscopy (ARPES). They proposed that the strong EPC and thus the large renormalized DOS could account for the highest superconducting transition temperature T_C observed in 2M-WS₂ (8.8 K) among all transition metal dichalcogenides. They tried further to explain the exotic properties like strange metal behavior in transport properties by using the phonon drag effect.

The paper provided two main indications in support of the strong EPC in this system. One is the nonmonotonic temperature dependence of the phonon linewidths; the other is the appearance of kinks in the dispersion of the low-energy electronic state measured by ARPES. However, both pieces of evidence have their own weakness.

Authors' response:

We thank the Referee for his/her pertinent reviewing. We greatly appreciate the Referee for acknowledging our experimental efforts on the temperature dependence of the phonon linewidths measured by Raman spectroscopy and the kinks in the dispersion of the low-energy electronic state measured by ARPES, which support the strong EPC in this system. In the following, we will provide a point-to-point response to the questions raised by the Referee. We hope our arguments can relieve his/her concerns about the weakness of our experimental evidence.

1. The interpretation of the Raman peak line width heavily depends on the model in eq. (1). But is that model the only way to interpret the data? The authors did not discuss the physical insight of the extra terms in eq. (1), corresponding to electron-phonon coupling. How reliable is this attribution?

Authors' response:

We thank the Referee for his/her comments on the interpretation of Raman measurements.

For materials with weak electron-phonon coupling, the frequencies (ω_{ph}) and linewidths (Γ) would show a general trend of phonon softening and broadening with increasing temperature, which can be well described by the model of an anharmonic decay assuming a symmetric decay of the optical phonon into two acoustic phonons [see Fig. 2b and *Phys. Rev.* 148, 845 (1966)]. A deviation from this model, especially the nonmonotonic behavior, suggests that additional scattering mechanisms need to be taken into consideration. Since 2M-WS₂ is nonmagnetic, the additional scattering channel can stem from electron-phonon interaction as electronic states match with the energy scale (\sim meV, \sim 100 K).

The extra electron-phonon coupling term in equation 1 has also been introduced to interpret phonon linewidths of NbGe₂ [*Nat. Commun.* 12, 5292 (2021)] and WP₂ [*Phys. Rev. X* 11, 011017 (2021)], which show similar temperature-dependent behaviors as 2M-WS₂. In the revised manuscript, a discussion on the physical insight of this term is included. As a phenomenological model illustrated in Fig. 2c, the temperature dependence of phonon-electron scattering must obey a Fermi-Dirac (instead of Bose-Einstein) function.

Regarding the reliability of the fitted linewidths from phonon-electron scattering, the Referee can also refer to our response to Question (2) of Reviewer #1.

2. The kinks in the dispersion curves in Fig. 3 and Fig. S3 appeared very poorly resolved. It is not clear whether argument made upon this is reliable.

Authors' response:

We thank the Referee for his/her comment.

We have checked the raw ARPES data of Fig. 3c. We believe our data is of high quality as evidenced by the decent signal-to-noise ratio and sharp band dispersions near the Fermi level (see the stacking MDC plots in Fig. R3a). One can clearly observe the kink features from the stacking MDC plots at around $E_F - 10$ meV and $E_F - 30$ meV from both the band dispersion (also see Fig. R3c) and the peak widths. Please note that the significantly broadened and incoherent peak profiles below $E_F - 30$ meV are typical spectral features due to many-particle interactions, which are **not** indications of poor data quality. Similar features have been commonly observed in cuprates (e.g., Bi2212 in Fig. R3e).

Fig. R3. Kink features of 2M-WS₂, Cs(V_{0.93}Nb_{0.07})₃Sb₅ and Bi2212. **a-c**, Kink features of 2M-WS₂. **a**, Stacking MDC plots. **b**, Same as Fig. 3c. **c**, Same as Fig. 3d,f. **d**, Kink features of Cs(V_{0.93}Nb_{0.07})₃Sb₅, adopted from Fig. 4 in *Nat. Commun.* 14, 1945 (2023). **e**, Kink features of Bi2212, adopted from Fig. 2 in *Nat. Commun.* 11, 569 (2020).

Fig. R3d and e present the kinks of Cs(V_{0.93}Nb_{0.07})₃Sb₅ [*Nat. Commun.* 14, 1945 (2023)] and Bi2212 [*Nat. Commun.* 11, 569 (2020)]. Compared with other recent ARPES studies on the low-energy kink features, our data have comparable or better quality and resolution.

We are confident that the arguments made upon the ARPES results are reliable through solid self-energy analyses. As elaborated in the manuscript, the resolved kink features have been analyzed through both the real part (band dispersion renormalization) and the imaginary part (Lorentzian broadening) of the self-energy. These two approaches lead to consistent kink energies for two bands, which strongly substantiate the reliability of our data analysis.

3. As the authors found, the strongly renormalized electronic density of states indicate the presence of correlated many-body physics. They also proposed that both the renormalized DOS and strong EPC are the key ingredients for the relatively higher T_C in 2M-WS₂. While it is understandable that the origin of the strong correlations is

beyond the scope of the paper, it is certainly not EPC alone. Therefore, when the authors commented on the two recent papers (Refs. 20 and 21) using the electron-phonon coupling picture alone, the argument is clearly stretched.

Authors' response:

We greatly appreciate the Referee for his/her criticism.

We fully agree with the Referee that both electron-electron correlations and electron-phonon coupling can contribute to the unique electromagnetic properties of 2M-WS₂. Refs. 20 and 21 are two pioneer studies that reveal signatures of correlated many-body physics in 2M-WS₂. We would like to clarify that we have **no** intention to claim that the properties reported in these two recent papers result from electron-phonon coupling alone. On the other hand, since the origin of these reported properties is not fully understood in the literature with various potential mechanisms discussed, we believe it is worth mentioning in our manuscript that electron-phonon coupling might play an important role.

In the revised manuscript, we have rewritten the discussion regarding these two recent papers and addressed that many-body interactions including both electron-electron correlations and electron-phonon coupling should be considered. We believe that our study can provide enlightening insights into the understanding of these unique properties.

4. In Fig. 6 of Supplementary Material, a phonon drag model is used to fit the temperature dependence of resistivity and there is also a phonon drag dominated region in Hall effect. Such a fitting may not be reliable. In Fig. 6(b) the model fits the data well up to $T \sim 60$ K, while in Fig. 6(d), the model deviates from the data as $T > 40$ K. However, both fits produce similar T_0 value. In Fig. (a) and (c), I do not know how the phonon dominated region can be determined reliably. How is $T_{\text{crossover}}$ determined in Fig.6 (c)? Moreover, in the phonon drag model, how is the magnetic field dependence of Hall coefficient (R_H) obtained?

Authors' response:

We thank the Referee for his/her pertinent comments.

First, we would like to emphasize that the interpretation of the temperature-dependent transport properties based on the phonon-drag picture is **not** a conclusion/claim of our manuscript. As we have stated in the response to Question 3, we have proposed this alternative explanation for the transport measurements reported in the literature and we believe it is worth discussing.

The Referee mentioned the discrepancy between Fig. S6a,b and Fig. S6c,d. This can be attributed to sample variance [the experimental data of Fig. S6a,b are extracted from *Nat. Phys.* 19, 379 (2023), whereas the experimental data of Fig. S6c,d are measured on one of our samples]. The low-temperature resistivity data can be well fitted by the phonon-drag formula, although the temperature regions are slightly different ($T \lesssim 70 K$ in Fig. S6b and $T \lesssim 50 K$ in Fig. S6d), possibly due to sample-dependent origins (e.g., impurity concentrations, size effect, dislocations). However, these two samples produce similar T_0 values, which correspond to the intrinsic activation energy $k_B T_0$ of electron-phonon scattering in the phonon-drag model [*Phys. Rev. Lett.* 37, 1574 (1976)].

We would like to address that the phonon-drag-dominated temperature region is **not** a well-defined concept based on a sharp phase transition. Empirically, the temperature-dependent resistivity can fit into the phonon-drag model, whereas the Hall coefficient starts to show clear magnetic field dependence when $T < T_{\text{crossover}}$, as shown in Fig. S6c.

In this phonon-drag-dominated region, the electrons and phonons form coupled quasiparticles, and a joint flow-velocity emerges as a collective hydrodynamic variable. For the physical mechanisms of the field dependence of the Hall coefficient in the phonon-drag condition, the Referee can refer to this theoretical work [*Annals of Physics* 419, 168218 (2020)]. As the magnetic field B is introduced into the motion equation by Lorentz term [see equation 3.12], the Hall coefficient becomes field-dependent [see equation 3.16]. As a candidate interpretation of the transport results, the phonon-drag model requires further examination from theoretical studies considering the significant electron-phonon coupling based on the complicated 3D Fermi surfaces of 2M-WS₂, which however is out of the scope of current manuscript.

Therefore, we have merely presented the transport phenomena we observed, which are likely to be related to the electron-phonon coupling discussed in our main manuscript, deserving further discussion. However, this observation does not serve as our final conclusion, as it necessitates additional experimental validation.

5. The temperature dependence of R_H in 2M-WS₂ is very similar to that in high- T_C cuprates (especially in slightly underdoped region). In cuprates, the origin of such a behavior is still under debate, and many works believe it is associated with some exotic scenarios, for example, the two different relaxation times related to different electron degrees in a strongly correlated system. Can a simple phonon drag model explain the Hall effect in both cuprates and 2M-WS₂?

Authors' response:

We greatly appreciate the Referee for his/her enlightening and insightful comments.

The temperature dependence of transport properties in 2M-WS₂ shows similarity to that in high- T_C cuprates [e.g., *Phys. Rev. Lett.* 67, 2088 (1991); *Phys. Rev. B* 69, R6991(R) (1999); *Phys. Rev. B* 64, 214504 (2001)]. We are aware that the origin of such transport behavior is far from understood. Various theories are proposed, including the “two-scattering-time” model [*Phys. Rev. Lett.* 67, 2092 (1991); *Phys. Rev. Lett.* 76, 1324 (1996)], the 1D stripe transport model [*Phys. Rev. B* 64, 214504 (2001)], as well as phonon drag and multiband conduction [*Phys. Rev. Lett.* 66, 1098 (1991)].

Our manuscript mainly focuses on the spectral evidence (Raman and ARPES) of strong and mode-selective electron-phonon coupling in 2M-WS₂. As we have stated in the response to Questions 3 and 4, the phonon-drag model of the transport behavior is **not** a conclusion of our manuscript but an alternative interpretation for discussion. The origin of the unique transport behavior in 2M-WS₂ is still debatable and requires further investigation. Therefore, we are refrained from associating the transport properties of cuprates with the phonon-drag model before direct evidence is provided.

2M-WS₂ and high- T_C cuprates do share many similarities, such as the low-energy kinks in the electronic dispersions and the temperature dependence of transport behaviors. In the

revised manuscript, we have included a discussion regarding these intriguing similarities and propose that 2M-WS₂ might become an important reference compound in the study of unconventional high- T_C superconductors.

In summary, we greatly appreciate the Referee about these useful comments and suggestions, which help us better explore the physics in the system and elaborate the details of the experiment. We hope that we have properly answered the questions raised by the second Referee and our response can relieve his/her concerns. We believe that the timely and important discovery in our work merits the requirements of *Nature Communications* and sincerely hope the second Referee can recommend publication of our manuscript.

Reply to Reviewer #3:

Reviewer #3: In this paper, the authors have quantitatively evaluated the strength of electron-phonon coupling in the topological superconductor 2M-WS₂, which has a relatively high superconducting transition temperature T_c (~ 8.8 K), by complementary use of Raman scattering and ARPES to investigate the origin of its high T_c . The results are consistent with the previous study based on the Migdal-Eliashberg theory, and the authors have concluded that it can be explained by the large density of states at $N(E_F)$ and the strong electron-phonon coupling from the comparison with other transition-metal dichalcogenides.

Authors' response:

We greatly appreciate the Referee for his/her pertinent reviewing. We thank the Referee for recognizing the main conclusion of our manuscript that the relatively high superconducting transition temperature T_c (~ 8.8 K) can be explained by the large density of states at $N(E_F)$ and the strong electron-phonon coupling by complementary use of Raman scattering and ARPES.

It is a very interesting result, and the paper basically seems to be worth being published in a high impact journal such as *Nature Communications*.

However, I have several concerns that I would like the authors to address before accepting the paper, and if possible, improve the manuscript.

Authors' response:

We are greatly grateful that the Referee found our result interesting and acknowledged that our paper basically seems to be worth being published in a high impact journal such as *Nature Communications*.

We'll provide a point-by-point response to the Referee's comments as follows, and we expect it can relieve his/her concerns.

First, I would like to see a little more detail on the fitting using Eq. (1) in line 133.

In the caption of Supplementary Fig. 2, it says “Upper panel: Fitted temperature-dependent phonon energy of 8 observed Raman-active phonon modes”. What kind of the fitting was performed here? The fitting to the Lorentzian? This should be clarified for the caption of upper panels.

Authors’ response:

The temperature dependence of the Raman linewidths is fitted based on Eq. (1) including 4 fitting parameters (Γ_0 , Γ_{ph-ph} , Γ_{ph-e} , and ω_e , please note that ω_0 is the optical phonon energy at zero temperature and we set it to the Raman measured frequency at the lowest temperature $\omega_{T=5K}$). As also suggested by Reviewer #1 in Question 2, we have presented the complete results of fitting parameters in the revised Supplementary Material.

In Fig. S2, the temperature-dependent Raman peak profiles are fitted to the Lorentzian function as $I(\omega) = \frac{I_0}{1 + \left(\frac{\omega - \omega_{ph}}{\Gamma_{ph}}\right)^2}$, where ω_{ph} is the phonon energy and Γ_{ph} is the phonon linewidth (half-width-at-half-maximum). We thank the Referee for his/her suggestion and we have clarified it in the revised caption of Fig. S2.

In line 131, it says “the observed nonmonotonic temperature dependent linewidths can be perfectly fitted by the following formula”, but how was ω_e in Eq. (1) treated in the fitting process? Treated as a fitting parameter? Wouldn’t it be more reasonable to multiply $N(\omega_e)$ or $N(\omega_e + \omega_0)$ and integrate? For ω_0 , was the phonon energy shown in the upper panels of Supplementary Fig. 2 used for each mode?

Authors’ response:

As mentioned in the response above, ω_e is one of the four fitting parameters in Eq. (1), whereas ω_0 is the optical phonon energy at zero temperature and we set it to the Raman measured frequency at the lowest temperature $\omega_{T=5K}$.

We agree with the Referee that the phonon-electron scattering term [the third term in Eq. (1)] should be proportional to $n_F(\omega_e, T) - n_F(\omega_e + \omega_0, T) = N(\omega_e)f(\omega_e, T) - N(\omega_e + \omega_0)f(\omega_e + \omega_0, T)$, where $N(\omega)$ is the electronic DOS and $f(\omega, T) = \frac{1}{\exp\frac{\omega}{k_b T} + 1}$ is the

Fermi-Dirac distribution function. However, in the energy region of interest (as ω_e and $\omega_e + \omega_0$ roughly fall into the energy window of $E_F \pm 50 \text{ meV}$), the Fermi-Dirac distribution functions in the temperature region of $\sim 100 \text{ K}$ ($\sim 8 \text{ meV}$) show much more significant energy dependence compared to that of the electronic DOS. Therefore, it is a reasonable approximation to treat $N(\omega)$ as a constant in the fitting process. Furthermore, although Eq. (1) is entirely phenomenological and does not rely on the results from *ab initio* calculations, the fitting results are satisfactory. Similar fitting processes have been applied in other materials, such as NbGe₂ [*Nat. Commun.* 12, 5292 (2021)] and WP₂ [*Phys. Rev. X* 11, 011017 (2021)].

We thank the Referee for this insightful comment and we have included the above arguments in the revised manuscript.

In line 147, it says $\lambda_{e-ph}^{Raman} = 0.32$ according to Eq (2), because both the phonon line width Γ_i and phonon energy ω_i seems to show temperature dependence, it is assumed that λ_{e-ph}^{Raman} also shows temperature dependence. Were the values extracted from the Raman measurements at the lowest temperature 5 K used? Or for the phonon line width Γ_i was Γ_{ph-e} obtained by the fitting using Eq (1) used for each mode?

In Table I, ω_0^{Raman} is stated as the value extracted from the Raman measurements at the lowest measurement temperature 5 K, but is the linewidth Γ_{ph-e}^i in Table I not the value extracted from the Lorentzian fitting of the Raman profile at 5 K? Or is it a value of Γ_{ph-e} obtained by fitting using Eq. (1) for each mode?

The authors should clarify how the values of Γ_i and ω_i were obtained to deduce $\lambda_{e-ph}^{Raman} = 0.32$ using Eq. (2).

Authors' response:

We thank the Referee for pointing out this confusion.

In Eq. (2), the total electron-phonon coupling strength is the summation of the electron-phonon coupling of each mode, which is associated with the phonon linewidth due to the

phonon-electron scattering. Experimentally, the value of the phonon energy ω_i is extracted from the Raman measurements at the lowest temperature (5 K) by fitting to a Lorentzian function; the value of Γ_i is Γ_{ph-e}^i obtained by the fitting using Eq. (1) for each mode, as presented in Table 1.

To avoid this confusion in the revised manuscript, we have modified the notations in Eq. (2) and clarified the definition of each quantity in the main text.

In Eq. (3) of line 184, it says $\lambda_{e-ph} \approx 1.1$, but again, it has not been clearly stated how the values used for ω_{ph}^i and $(\Delta\text{Im}\Sigma)^i$ were obtained. Was ω_0^{Raman} used for ω_{ph}^i ? For $(\Delta\text{Im}\Sigma)^i$, it seems to have been shown in Fig. 3g how to evaluate $(\Delta\text{Im}\Sigma)^i$, but it should be mentioned in the figure caption and main text.

Authors' response:

We thank the Referee for his/her comments.

By using Eq. (3), we have estimated the electron-phonon coupling from the imaginary part of the self-energy extracted from ARPES. Here, ω_i are the phonon energies which correspond to the kink energies observed by ARPES, with $\omega_1 \approx 10 \text{ meV}$ and $\omega_2 \approx 30 \text{ meV}$. $(\Delta\text{Im}\Sigma)^i$ are the steplike increases of the imaginary part of the self-energy between two adjacent plateaus, with $(\Delta\text{Im}\Sigma)^1 \approx 9.7 \text{ meV}$ and $(\Delta\text{Im}\Sigma)^2 \approx 24.9 \text{ meV}$, as shown in Fig. 3g.

In the revised manuscript, we have stated in the main text and in the caption how these values are experimentally obtained. We thank again for the comments from the Referee which improve the rigidity of our manuscript.

In line 174, it says $\lambda_{\text{tot.}}^{\text{DOS}} = 1.9$, but according to $N_0 = 1.38$ in line 105 and $N(E_F) = 3.81$ in line 107, $N(E_F)/N_0 - 1$ should become 1.76. I think it may be reasonably consistent, but "matches perfectly" seems to be an overstatement.

Authors' response:

We thank the Referee for pointing out this error.

We agree with the Referee and we have modified accordingly in the revised manuscript. The total renormalization factor regarding DOS at E_F is $\lambda_{tot.}^{DOS} = \frac{N(E_F)}{N_0} - 1 = 1.76$. This value ($\lambda_{tot.}^{DOS} = 1.76$) is nicely consistent with that deduced from the renormalization of the Fermi velocity ($\lambda_{tot.}^{v_F} = 1.9$).

Finally, while Fig. 4b is a very interesting plot, I have a concern that it does not include 2M-WSe₂, which seems to be the most suitable reference material. According to the co-authors of this paper, 2M-WSe₂ does not seem to exhibit superconductivity at ambient pressure, but it does under high pressure, and the ARPES measurements have already been performed; is there any valid reason not to perform similar measurements and analyses for 2M-WSe₂ for direct comparison? If there is no valid reason and it is feasible to perform them, the results should be included to validate the authors' conclusions.

Authors' response:

We thank the Referee for his insightful comments and enlightening suggestions.

We agree with the Referee that 2M-WSe₂ might be the most suitable reference material for 2M-WSe₂. They share the same crystal structure but exhibit distinct electronic structure and transport properties. 2M-WSe₂ is a weak topological insulator with no topological surface states on the naturally cleaved (100) surface [*Adv. Mater.* 35, 230027 (2023)]. It does not exhibit superconductivity at ambient pressure, but it does under high pressure [*Journal of Materials Chemistry C* 7, 8551 (2019)].

Although being a suitable reference material, we exclude 2M-WSe₂ in Fig. 4b because the renormalized DOS and electron-phonon coupling strength λ_{e-ph} are still unavailable from experiments. As we have elaborated in the caption of Fig. 4b and Supplementary Table 2, the renormalized DOS at E_F $N^*(E_F)$ is extracted from the Sommerfeld coefficient derived from measurements of temperature dependence of specific heat and the electron-phonon coupling strength λ_{e-ph} is derived by McMillan formula based on the superconducting transition temperature (T_C) and the Debye temperature (Θ_D). However, no measurement on the specific heat of 2M-WSe₂ has been found and it exhibits no superconducting transition.

Fig. R4. Band structure characterization of 2M-WSe₂. **a**, The calculated noninteracting band structure projected on the (100) surface (left panel) and corresponding DOS (right panel). **b**, Synchrotron-ARPES measured band dispersion along the $\bar{Y} - \bar{\Gamma} - \bar{Y}$ direction. **c**, Laser-ARPES measured band dispersion in the momentum-energy region as indicated by the gray rectangle in **b**. **d**, Corresponding band structure calculation in the same momentum-energy region as in **c**.

As suggested by the Referee, we agree that results of similar measurements and analyses on 2M-WSe₂ using *ab initio* band structure calculation, ARPES, and Raman spectroscopy can provide a direct comparison with the results of 2M-WS₂, which should be helpful to validate our conclusion. As shown in Fig. R4a, the calculated noninteracting DOS of 2M-WSe₂ at E_F is 0.79 states/eV/f.u., which is significantly lower than that of 2M-WS₂ (1.38 states/eV/f.u.). Both synchrotron-ARPES (Fig. R4b) and laser-ARPES (Fig. R4c) show no signature of kinks in the low-energy electronic dispersions. The absence of the kink features can be an indication of the relatively low EPC in 2M-WSe₂. However, we cannot rule out the scenario that the kink features are strongly interfered by the bulk continuum, as shown in Fig. R4d.

Fig. R5. Raman spectroscopy characterization of 2M-WSe₂. **a**, Temperature-dependent Raman spectra from 5 to 300 K. **b-h**, Upper panel: Fitted temperature-dependent phonon energies of 7 observed modes based on Lorentzian functions. Lower panel: Fitted temperature-dependent phonon linewidths (half-width-at-half-maximum of the Lorentzian profile) of 7 observed modes. The fitted results are interpreted by either phonon-electron scattering (blue shaded area in **b**) or phonon-phonon scattering (red shaded area in **c-h**) based on Eq. (1) in the main text. The fitted results from these two scattering mechanisms are upshifted by a constant background Γ_0 (as indicated by black arrows) for a better illustration.

Similar temperature-dependent Raman spectroscopy characterization of 2M-WSe₂ has been performed, as shown in Fig. R5. 7 phonon modes have been observed (Fig. R5a), consistent with previous reports [*Journal of Materials Chemistry C* 7, 8551 (2019)]. In contrast to the results of 2M-WS₂, $B_g(1)$ is the only mode showing decreasing phonon linewidth with increasing temperature (Fig. R5b), whereas other 6 modes show conventional increasing linewidth with increasing temperature (Fig. R5c-h). The linewidths of these 6 phonon modes can be well fitted by the phonon-phonon scattering mechanism without the contribution from the phonon-electron scattering mechanism, as shown in the lower panel of Fig. R5c-h. For $B_g(1)$ mode, the fitted Γ_{ph-e} is 0.09 meV, which is significantly smaller than that of 2M-WS₂ (0.41, please refer to Table 1 in the main text).

In conclusion, ARPES and Raman spectroscopy characterizations of 2M-WSe₂ show no experimental signature of strong electron-phonon coupling, in great contrast to the results of

2M-WS₂. We thus suggest that the difference of electron-phonon coupling strengths between 2M-WS₂ and 2M-WSe₂ leads to their distinct superconducting behaviors.

Following the Referee's suggestion, we have included a brief discussion on the comparison between 2M-WS₂ and 2M-WSe₂ in the revised manuscript and we have included Fig. R4 and Fig. R5 in the revised Supplementary Material to validate our conclusions.

If the authors can appropriately answer to these comments, I would then like to recommend this paper for publication in *Nature Communications*.

Authors' response:

We thank the Referee again for his/her insightful comments, which are greatly helpful in improving the quality of our manuscript. We hope we have properly answered to these comments and that our revised manuscript can meet the standard for publication in *Nature Communications*.

REVIEWER COMMENTS

Reviewer #1 (Remarks to the Author):

I have reviewed the responses from the authors and found that there still exist severe problems regarding the Raman spectra fitting data and the updated electron-phonon coupling (EPC) constant deduced from Raman phonon linewidth, which affect the reliability of this work, as described below:

(1) By reshaping the equation 2, the authors reevaluate the total EPC constant to be $\lambda = 3.53$ by summing over all the measured Raman-active optical phonon modes and attributed this obvious overestimation (over 0.8) to the strong electron correlation in 2M-WS2. Such a substantial inaccuracy of Raman measurements is rarely seen in literature. For instance, even in the well-known high-temperature superconductor YBa2Cu3O7 and moderately high-temperature superconductors M3C60 [PRB 42, 2692(R) (1990) and PRB 48, 8412 (1994)], Raman experiments reported only total EPC of around 1.0 and 0.5-0.6, respectively, with the latter being in accord with theoretical calculations. Considering that there is no magnetism and no direct evidence of strong electron correlation in 2M-WS2 (the observed kinks in ARPES indicated strong electron-phonon coupling, but not electron correlation), the equation 1 incorporating both phonon-phonon and phonon-electron scattering channels is expected to be appropriate for describing the total linewidth and thus should result in a good estimate of the total EPC based on the linewidth from phonon-electron scattering. Hence, the failure of reproducing the total EPC constant from Raman experiments here is hard to understand.

(2) Does this failure have to do with the use of different background parameters Γ_0 for different phonon modes, which is clearly seen from the updated Table S1 where even zero Γ_0 values are adopted for several modes. As a background parameter, why Γ_0 is so different for the varying phonon modes? Is there any physical interpretation?

(3) The authors derived the phonon linewidth based on Lorentzian functions. However, in previous works [PRB 37, 5920 (1988), PRB 42, 2692(R) (1990) and PRB 48, 8412 (1994)], asymmetric line shape of the Raman spectra suggests electron-phonon interaction for the phonon modes investigated, and phonon linewidths were derived based on the Breit-Wigner-Fano (BWF) resonance line shape. Thus it would be necessary to check whether the Fano line shape can fit the Raman peaks better than the Lorentzian line shape and how the total EPC changes correspondingly.

(4) For completeness, is there any reason for not reporting the additional data for 2M-WS2 in Fig. R2 about its Fermi velocity renormalization along different momentum directions and the relevant discussion in the revised manuscript? Moreover, it should be noted that anisotropic multiband superconductivity in 2M-WS2 has been confirmed in a recent detailed experimental study of the temperature-dependent London penetration depth [see PHYSICAL REVIEW RESEARCH 6, 013124 (2024)].

Reviewer #2 (Remarks to the Author):

Most of my concerns have been addressed in the revisions and the manuscript has been improved. I think it can be accepted for publication.

Reviewer #3 (Remarks to the Author):

The authors have adequately addressed all of the reviewers' concerns, and the quality of the paper seems to have been greatly improved by this revision. In particular, the addition of the 2M-WSe₂ data seems to have further clarified the importance of electron-phonon coupling in 2M-WS₂. I would like to express my respect for the authors' efforts for this revision. Therefore, as I wrote in my previous report, I would like to recommend this paper for publication in Nature Communications.

Reply to Reviewer #1:

I have reviewed the responses from the authors and found that there still exist severe problems regarding the Raman spectra fitting data and the updated electron-phonon coupling (EPC) constant deduced from Raman phonon linewidth, which affect the reliability of this work, as described below:

Authors' response:

We greatly appreciate the Reviewer for his/her pertinent reviewing and we understand the Reviewer's concern regarding our data analysis of Raman measurements. The Reviewer's comments are insightful and considerably help us further improve the rigorousness of our manuscript.

Before our point-by-point response addressed by the Reviewer, we would like to emphasize that the exact value of electron-phonon coupling (EPC) constant deduced from Raman phonon linewidths is **not** the main conclusion of our work. We must admit that the data analysis of Raman measurements can only lead to a rough estimation of the EPC constant with significant uncertainty. **Such an ambiguity is common in literature** (as we will elaborate in the following response). However, we believe our main conclusion on the relatively strong EPC in 2M-WS₂ is solid and supported by the observation of the anomalous nonmonotonic temperature dependence of Raman phonon linewidths.

We believe our point-by-point response can dispel the Reviewer's concern. We sincerely hope the Reviewer can reconsider his/her evaluation and recommend publication of our work.

(1) By reshaping the equation 2, the authors reevaluate the total EPC constant to be $\lambda = 3.53$ by summing over all the measured Raman-active optical phonon modes and attributed this obvious overestimation (over 0.8) to the strong electron correlation in 2M-WS₂. Such a substantial inaccuracy of Raman measurements is rarely seen in literature. For instance, even in the well-known high-temperature superconductor YBa₂Cu₃O₇ and moderately high-temperature superconductors M₃C₆₀ [PRB 42, 2692(R) (1990) and PRB 48, 8412 (1994)], Raman experiments reported only total EPC of around 1.0 and 0.5-0.6,

respectively, with the latter being in accord with theoretical calculations. Considering that there is no magnetism and no direct evidence of strong electron correlation in $2M\text{-WS}_2$ (the observed kinks in ARPES indicated strong electron-phonon coupling, but not electron correlation), the equation 1 incorporating both phonon-phonon and phonon-electron scattering channels is expected to be appropriate for describing the total linewidth and thus should result in a good estimate of the total EPC based on the linewidth from phonon-electron scattering. Hence, the failure of reproducing the total EPC constant from Raman experiments here is hard to understand.

Authors' response:

We understand the Reviewer's concern about "the failure of reproducing the total EPC constant from Raman experiments". The Reviewer mentioned that "such a substantial inaccuracy of Raman measurements is rarely seen in literature" and provided two studies on the cuprate superconductor $\text{YBa}_2\text{Cu}_3\text{O}_7$ and M_3C_{60} [PRB 42, 2692(R) (1990) and PRB 48, 8412 (1993)], which report seemingly reasonable values of EPC constants. However, by careful literature reviewing (including these two studies), **we respectfully argue that analysis of the Raman phonon linewidths can only provide a very rough estimation of the EPC constant with significant uncertainty.**

For the study on the cuprate superconductor $\text{YBa}_2\text{Cu}_3\text{O}_7$ [PRB 42, 2692(R) (1990)], the estimation of $\lambda \approx 1$ is deduced from the **calculated** phonon linewidths and partial EPC constants of five Raman-active A_g modes. Although this theoretical study is consistent with several experimental observations, such as the magnitude of the softenings and hardenings of the A_g Raman modes below T_C , the estimation of the EPC constant is a **calculated result** of the local-density approximation but **not a result deduced from Raman experiments.**

Furthermore, the calculated estimation of $\lambda \approx 1$ has an extremely large uncertainty. Relevant excerpt (a paragraph on page 2694) and Figure 1 of the literature are shown below:

"Figure 1 shows $\gamma_{\nu q}/\omega_\nu$ and $\lambda_{\nu q}$ as functions of q_{ab} for $q_c = 0$ and averaged over the angles in the ab plane.....By taking the small- q average of the $\lambda_{\nu q}$ curves, by

averaging over the five A_g modes, and by multiplying by the total number of modes (39), one may estimate a total λ of order 1.”

One can see that the estimation is very rough and it neglects the mode dependence of the EPC since the total λ of 39 phonon modes is deduced by averaging the 5 A_g modes. However, the mode dependence is obvious even in these 5 A_g modes. As shown in Fig. 1 of *Phys. Rev. B* 42, 2692(R) (1990), the averaged $39\lambda_\nu$ is around 0.5 for 500 cm^{-1} O(4) mode, but around 4.0 for 110 cm^{-1} Ba mode. Therefore, **the EPC constant is estimated to be of order 1 and might range from 0.5 to 5.**

For the study on M_3C_{60} [PRB 48, 8412 (1993)], the values of λ are estimated to be 0.6 and 0.5 for K_3C_{60} and Rb_3C_{60} by Raman scattering. Although the data analysis in this study is based on the same formula as in our work [Eq. (1) of PRB 48, 8412 (1993) is identical to Eq. (2) of our manuscript], the phonon linewidth broadening due to the electron-phonon interaction [denoted as $\Delta\Gamma_i$ in PRB 48, 8412 (1993) and Γ_{ph-e}^i in our manuscript] are determined differently. In the literature, $\Delta\Gamma_i$ is the difference in the linewidth of corresponding phonon modes in pristine C_{60} and M_3C_{60} at 300 K, while in our manuscript, Γ_{ph-e}^i is fitted from temperature-dependent Raman phonon linewidths.

Apart from the difference in methodology, we would like to emphasize that the estimated EPC constants also have significant uncertainties. They arise from **a rough estimation of the density of states at the Fermi level $N(E_F)$** , as excerpted below [page 8415-8416 of PRB 48, 8412 (1993)]:

“..... $N(\epsilon_f) = 15$ states/eV spin C_{60} , where $N(\epsilon_f)$ represents a rough average over several experimental values and theoretical calculations: 1-2 states/eV spin C_{60} from photoemission studies, 6-20 states/eV spin C_{60} from band calculations, and 10-16 states/eV spin C_{60} from NMR studies.”

One can see that **the uncertainty in $N(E_F)$ (1-20) must result in a large variation of the EPC constant over one order of magnitude.**

From the discussion of these two typical studies on the estimation of EPC constants, **we argue that Raman scattering measurement has obvious limitations in the accurate**

estimation of EPC constant. In general, only the order of magnitude of the EPC constant can be reliably obtained. In this sense, the EPC constant of 2M-WS₂ deduced from Raman experiments (3.53) and *ab initio* calculation (0.8) are consistent and are both of order 1.

We agree with the Reviewer that it is a bit handwaving to attribute the overestimation of the EPC constant to the strong electron correlation in 2M-WS₂. After the literature review, we believe that the discrepancy is more likely to be attributed to the intrinsic uncertainty in the data analysis of Raman phonon linewidths. The origin of the variance however can be complicated, such as inaccurate fitting models, unstable fitting procedures, or improper choices of background, which we will further elaborate in the following response.

We would like to thank the Reviewer again for his/her enlightening comments. To improve the rigorousness of our manuscript, we have revised the discussion on the inaccuracy of the EPC constants deduced by Raman measurements. The revision has further improved the rigorousness of our work and would not affect the main conclusion of our study.

(2) Does this failure have to do with the use of different background parameters Γ_0 for different phonon modes, which is clearly seen from the updated Table S1 where even zero Γ_0 values are adopted for several modes. As a background parameter, why Γ_0 is so different for the varying phonon modes? Is there any physical interpretation?

Authors' response:

We greatly thank the Reviewer for this constructive comment.

Based on Eq. (1) in the main text, we agree with the Reviewer that improper choices of the background parameters Γ_0 can lead to unreliable fitting results of Γ_{ph-e} . For example, an underestimation of Γ_0 may result in an overestimation of Γ_{ph-e} . However, we are **not** aware of any physical reasons that guarantee identical Γ_0 for different modes. It can be commonly seen in literature that Γ_0 are different for varying phonon modes. For instance, in the study of Raman-active phonon modes of MoS₂ thin films [PRB, 101, 205302 (2020)], Γ_0 (FWHM at zero temperature limit) of E_{2g} and A_{1g} are around 1.4 cm⁻¹ and 2.4 cm⁻¹, respectively, for the single-layer MoS₂ [please refer to Fig. 3 of PRB, 101, 205302 (2020)].

The physical origins of different Γ_0 can be complicated, such as phonon dispersions, experiment geometries, and various types of defects, however beyond the scope of current manuscript.

In our fitting procedure, Γ_0 is one of the four **free** fitting parameters (Γ_0 , Γ_{ph-ph} , Γ_{ph-e} , and ω_e). This results in satisfying fitting results of the temperature-dependent phonon linewidths (see Fig. 2e-g in the main text and Supplementary Fig. 2). However, to improve the stability of the fitting procedure, less free fitting parameters are favorable.

For this purpose, we classify the 8 Raman-active phonon modes into two categories. The first category includes $B_g(1)$, $A_g(1)$, $A_g(2)$, and $A_g(3)/B_g(3)$ showing the nonmonotonic temperature-dependent behavior (see Fig. R6a-c and e). The second category includes $B_g(2)$, $A_g(4)$, $A_g(5)$, and $A_g(6)$ showing the conventional monotonic increase of phonon linewidths with temperature (see Fig. R6d and f-h). Phonon-electron scattering and phonon-phonon scattering dominate in these two categories, respectively. One can see that contributions from one mechanism of phonon-electron or phonon-phonon scattering can lead to decent fitting results for the 8 Raman-active phonon modes, as shown in Fig. R6.

Fig. R6. Fitting results based on phonon-electron or phonon-phonon scattering for the 8 Raman-active phonon modes of 2M-WS₂. Constraints of Γ_0 are also applied.

In this fitting procedure, constraints of Γ_0 are also applied. Based on Eq. (1) in the main text, one can see that the linewidth broadenings due to phonon-phonon and phonon-electron

scattering converge to zero at zero temperature when $\omega_e > 0$. Therefore, we set $\Gamma_0 = \Gamma(T = 5 \text{ K})$ for all modes except for $A_g(1)$ and $A_g(3)/B_g(1)$. For these two modes, the zero-temperature limit is inapplicable for $T = 5 \text{ K}$ since ω_e and $k_B T$ are comparable. Γ_0 of all 8 Raman-active modes are of the same order of magnitude (0.23~0.75 meV) and zero Γ_0 are avoided.

The reevaluated total EPC constant based on this fitting procedure is $\lambda_{e-ph} = 2.38$. This value is smaller than the previous estimation (3.52) but still larger than the theoretical value (0.8), however, they are all of order 1. The inaccuracy of the EPC constant deduced from Raman phonon linewidths has been elaborated in the response to Question (1) and it has complicated physical origins. In our case, we argue that Eq. (1) in the main text is not a first-principles result but merely based on an extremely simple and rough phenomenological model. For the phonon-phonon scattering, only the lowest order of anharmonicity is included. For the phonon-electron scattering, Eq. (1) does not rely on phononic or electronic structures, as well as their temperature dependence. Therefore, the estimation of Γ_{ph-e} by fitting the temperature-dependent Raman phonon linewidths to Eq. (1) has limited accuracy.

As enlightened by the Reviewer, we have included the above discussion in the revised manuscript, regarding the possible origins of the inaccurate estimation of the EPC constant. Fig. R6 is also included in the revised Supplementary Material. **We would like to emphasize again that our estimated EPC constant is of order 1, which is consistent with the *ab initio* calculation (0.8).** The exact value of the EPC constant is **not** our main conclusion, whereas the experimental signatures of the nonmonotonic temperature-dependent Raman phonon linewidths are unambiguously observed, indicating strong EPC in 2M-WS₂.

(3) The authors derived the phonon linewidth based on Lorentzian functions. However, in previous works [PRB 37, 5920 (1988), PRB 42, 2692(R) (1990) and PRB 48, 8412 (1994)], asymmetric line shape of the Raman spectra suggests electron-phonon interaction for the phonon modes investigated, and phonon linewidths were derived based on the Breit-Wigner-Fano (BWF) resonance line shape. Thus it would be necessary

to check whether the Fano line shape can fit the Raman peaks better than the Lorentzian line shape and how the total EPC changes correspondingly.

Authors' response:

We greatly thank the Reviewer for his/her suggestion.

As shown in Fig. R7, we have checked the fitting results by using the Fano line shape for all 8 Raman-active phonon modes. One can see that Lorentzian and Fano line shape fittings result in perfectly consistent linewidths for all 8 modes. The fitted frequencies show small discrepancies (< 0.5 meV) only for $A_g(1)$ in the whole temperature range (5~300 K) and $A_g(6)$ in the high-temperature region ($T > 100$ K).

Fig. R7. Comparison of fitted phonon frequencies and linewidths by using Lorentzian and Fano line shapes for all observed 8 Raman-active modes.

The small differences in frequencies and linewidths between Lorentzian and Fano fittings should have negligible influence ($< 1\%$) in the estimation of the total EPC constant, based on Eq. (2) in the main text.

The most pronounced asymmetric line profile has been observed in $A_g(2)$ mode, as shown in Fig. R8. As asymmetric Fano line shape is an experimental indication for strong EPC, this observation is consistent with our fitting results showing in Fig. R6c that $A_g(2)$ mode exhibits the strongest partial EPC constant (1.06) among all 8 Raman-active modes.

Fig. R8. Lorentzian (a,c) and Fano (b,d) line profile fittings of Raman spectra of phonon mode $A_g(2)$ measured at 5 K (a,b) and 300 K (c,d).

Fig. R7 and Fig. R8 are included in the revised Supplementary Material.

(4) For completeness, is there any reason for not reporting the additional data for 2M-WS₂ in Fig. R2 about its Fermi velocity renormalization along different momentum directions and the relevant discussion in the revised manuscript? Moreover, it should be noted that anisotropic multiband superconductivity in 2M-WS₂ has been confirmed in a recent detailed experimental study of the temperature-dependent London penetration depth [see PHYSICAL REVIEW RESEARCH 6, 013124 (2024)].

Authors' response:

We have provided convincing reasons in our previous response for not reporting the additional data for 2M-WS₂ in Fig. R2 about its Fermi velocity renormalization along different momentum directions. The relevant excerpt is shown below:

*“However, we believe the variance is **not** intrinsic and does **not** evidence momentum dependence of the many-body interaction for the following reasons. The above analysis approach can be inaccurate considering the mixture of the surface and bulk states.....The*

uncertainty of bulk dispersions due to k_z broadening can lead to unreliable results and significant overestimation of the coupling strength.”

However, as suggested by the Reviewer, we have included Fig. R2 (of our previous response) and relevant discussion in the revised Supplementary Material for completeness.

We are aware of studies reporting anisotropic superconductivity in 2M-WS₂ [*Nano Lett.* **21**, 709 (2021); *Nat. Phys.* **15**, 1046 (2020); *Phys. Rev. Res.* **6**, 013124 (2024)]. Although it might not be perfectly isotropic, we argue that anisotropy in the EPC of 2M-WS₂ can be weak. This is supported by the theoretical study [*Nano Lett.* **21**, 709 (2021)]. Isotropic Eliashberg spectral function $\alpha^2F(\omega)$ results in an EPC constant of 0.79, which is consistent with experimental superconducting transition temperature [please refer to Fig. 1(c) of *Nano Lett.* **21**, 709 (2021)]. Furthermore, the difference between the calculated superconducting gaps using isotropic and anisotropic Migdal-Eliashberg theory is relatively small [~ 0.2 meV, please refer to Fig. 1(d) of *Nano Lett.* **21**, 709 (2021)].

For rigorousness, the discussion on the momentum dependence of many-body interactions has been updated in the revised manuscript.

With the consideration discussed above, we believe that our study is solid based on rigorous data analysis and our experimental discoveries are significant, and thus hope the Reviewer will reconsider his/her assessment and recommend our work for publication in *Nature Communications*.

Reply to Reviewer #2:

Most of my concerns have been addressed in the revisions and the manuscript has been improved. I think it can be accepted for publication.

Authors' response:

We appreciate that the Reviewer's concerns have been relieved and the improvement of our manuscript has been recognized. We greatly thank the Reviewer for his/her recommendation for publication.

Reply to Reviewer #3:

The authors have adequately addressed all of the reviewers' concerns, and the quality of the paper seems to have been greatly improved by this revision. In particular, the addition of the 2M-WSe₂ data seems to have further clarified the importance of electron-phonon coupling in 2M-WS₂. I would like to express my respect for the authors' efforts for this revision. Therefore, as I wrote in my previous report, I would like to recommend this paper for publication in *Nature Communications*.

Authors' response:

We are greatly grateful that our efforts for the revision have been appreciated by the Reviewer. We sincerely thank the Reviewer for his/her recommendation for publication in *Nature Communications*.

REVIEWERS' COMMENTS

Reviewer #1 (Remarks to the Author):

I appreciate the changes made in response to my previous report, and the current manuscript is definitely improved. I therefore recommend its publication in Nat. Commun.